# Contribution of open ocean to the nutrient and phytoplankton inventory in a semi-enclosed coastal sea

Qian Leng[1], Xinyu Guo[2], Junying Zhu[3], Akihiko Morimoto[2]

[1]Graduate School of Science and Engineering, Ehime University, 2-5 Bunkyo-cho, Matsuyama, Ehime 790-8577, Japan
[2]Center for Marine Environmental Studies, Ehime University, 2-5 Bunkyo-cho, Matsuyama, Ehime 790-8577, Japan
[3]State Key Laboratory of Marine Resource Utilization in South China Sea, Hainan University, Haikou 570228, China

*Correspondence to*: Xinyu Guo (guo.xinyu.mz@ehime-u.ac.jp)

**Abstract.** The semi-enclosed coastal seas serve as a transition zone between land and open ocean and their environments are therefore affected by both. The influences of land were noticed but that of the open ocean were usually neglected. The Seto Inland Sea (SIS), which is connected to the Pacific Ocean, is a typical representative of semi-enclosed seas. To quantitatively assess the inventory of nutrients originating from land and open ocean, and their supported phytoplankton in the SIS, we developed a three-dimensional coupled hydrodynamic-biogeochemical model and embedded a tracking technique in it. Model results showed that the open ocean contributes 61% and 46% to the annual inventory of dissolved inorganic nitrogen (DIN) and phytoplankton in the SIS, respectively. This proportion has apparent spatial variations: being highest near the boundary with the open ocean, decreasing from there towards the interior area of SIS, and being lowest in the nearshore areas. The open ocean imports 799 $mol\ s^{-1}$ of DIN to the SIS, 25% of which is consumed by biogeochemical processes, and 75% is delivered again to the open ocean. Such a large amount of oceanic nutrient input and its large contribution to the inventory of DIN and phytoplankton suggest the necessity to consider the impact of the open ocean variabilities in the management of land loading of nutrients for the semi-enclosed seas.

## 1 Introduction

The semi-enclosed coastal seas geographically refer to the seas surrounded by land and connected to the open ocean by one or more narrow entrances (Healy and Harada, 1991; Leppäkoski et al., 2009). These regions are highly productive due to their location at the transition zone between land and open ocean, which results in the receipt of nutrients from both terrestrial and oceanic sources (Bauer et al., 2013; Jickells, 1998). While it was previously believed that the narrow entrances limited the exchange between the semi-enclosed seas and the open ocean (Caddy, 2000; Statham, 2012), recent studies have highlighted the significant contribution of oceanic nutrients to the nutrient inventory in the semi-enclosed seas (Anderson et al., 2008; Mackas and Harrison, 1997; Townsend, 1998). Therefore, the role of oceanic nutrients cannot be disregarded in semi-enclosed seas. In fact, it has been known that the changes in terrestrial nutrients alone cannot fully explain ecological variations in these regions (Nakai et al., 2018).

Unlike terrestrial nutrients, which are susceptible to anthropogenic activities, oceanic nutrients are typically influenced by climate change, which affects ocean currents and other environmental variables, including nutrient concentrations in the open ocean (Jickells, 1998; Vermaat et al., 2008). The transport process of oceanic nutrients into the coastal seas depends on the water exchange between the semi-enclosed seas and the open ocean, so their import has a seasonality that is different from the terrestrial nutrients (Morimoto et al., 2022; Zhang et al., 2019). Because of its complex nature, the transport of oceanic nutrients

into the semi-enclosed seas and their role in these seas are not easily known. One example is the Seto Inland Sea (SIS), where the presence of oceanic nutrients has been recognized for a long time (Hayami et al., 2004; Yanagi and Ishii, 2004). However, their transport flux into the SIS as well as their contribution to the primary production there have not been documented.

    The impacts of an external source of nutrients are usually assessed by the amount of input into the coastal seas (Aoki et al., 2022), which actually reflects only its potential impact, not its real impact. One reason is that not all nutrients entering the

coastal seas are available for phytoplankton growth (Zhang et al., 2019). Thus, the evaluation of the input amount of oceanic or terrestrial nutrients is only the first step in understanding their effects in semi-enclosed seas. Their real effects should be evaluated by their inventory of nutrients and biological particles (phytoplankton, zooplankton, and detritus) in the semi-enclosed seas, which is associated with the material flows in biogeochemical cycling and the material exchange at the boundary between semi-enclosed seas and open ocean.

In-situ observations can often be insufficient to differentiate materials originating from different sources of nutrients. To address this issue, researchers have proposed to introduce the tracking modules into the coupled hydrodynamic-biogeochemical models to trace the nutrients from different sources (Kawamiya, 2001; Ménesguen et al., 2006). Recently, this type of model has demonstrated their performance in quantifying the contribution of riverine and oceanic nitrogen to hypoxia formation in the Northern Gulf of Mexico, where the oceanic nitrogen supports 16±2% of summer sediment oxygen

consumption (Große et al., 2019). Similar models have also been used to assess the contribution of different nutrient sources to primary production in the East China Sea, where the nutrients from the Kuroshio support up to 50% of primary production (Zhang et al., 2019).

    This study focuses on the SIS, a semi-enclosed sea that is connected to the Pacific Ocean. Heavy eutrophication occurred in the SIS during the rapid economic growth of the 1960s-1970s due to urbanization and concentrations of industry and population

around it. After that, Temporary Law for Environment Conservation in SIS and Special Law for Environment Conservation in SIS was made in 1973 and 1978, respectively. Based on the law, total amount controls of Chemical Oxygen Demand (COD), nitrogen, and phosphorus have been conducted since then. As a result of a long time of effort, the nutrient concentration in some areas of the SIS has largely decreased, which raised a social concern about the possibility of oligotrophication in the SIS (Yamamoto, 2003).

To initiate our understanding of such long-term change in the nutrient concentrations in the SIS, we use a coupled hydrodynamic-biogeochemical model with a tracking technique to quantitatively evaluate the inventory of materials originating from the open ocean, river, and sediment. In addition, we emphasize the material flows involved in biogeochemical

cycling and material exchange through the boundary with the open ocean. Through these analyses, we want to present an example to demonstrate the role of oceanic nutrients in semi-enclosed seas.

## 2 Methods

### 2.1 Study area

The SIS is the largest semi-enclosed coastal sea in western Japan. It has a surface area of approximately 23,203 km$^2$ and an average depth of 38 m (Fig. 1a). The sea is surrounded by three major islands, namely Honshu, Kyushu, and Shikoku, and is connected to the Pacific Ocean through two straits, the Bungo Channel and Kii Channel (Fig. 1b). During the summer, the cold and nutrient-rich Kuroshio subsurface water from the Pacific Ocean intrudes into the SIS through these two straits (Kobayashi and Fujiwara, 2009; Takashi et al., 2006; Guo et al., 2004). The SIS is surrounded by a highly industrialized area in Japan where more than 10 million people live. Consequently, the nutrients discharged from land are high (Yamamoto, 2003). Furthermore, the organic material in the sediment also releases nutrients (Tada et al., 2018; Yamamoto, 2003). Therefore, we consider three sources of nutrients for the SIS in this study.

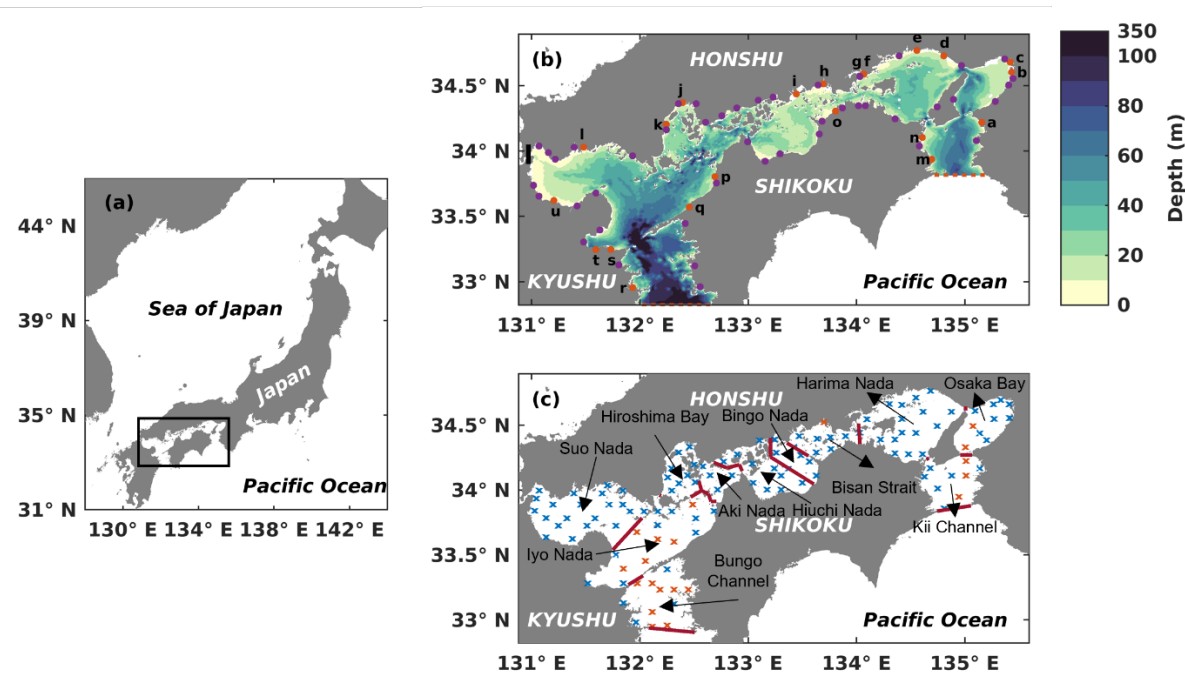

**Figure 1**. **(a)** Location of the Seto Inland Sea. **(b)** Bathymetry of the Seto Inland Sea. The solid orange circles with lowercase letters represent the locations of first-class rivers, and the solid purple circles represent the locations of second-class rivers. The names of 21 first-class rivers are shown in Table S2. The red dashed line in the west represents the open boundary between Bungo Channel and the open ocean, and the red dashed line in the east represents the open boundary between Kii Channel and the open ocean. The solid black line represents the location of Kanmon Strait. **(c)** Division of sub-regions and locations of in-

situ observation stations. Red solid lines represent dividing lines of different sub-regions. Blue crosses represent in-situ observation stations with a depth of less than 50 m, and orange crosses represent in-situ observation stations with a depth greater than 50 m.

## 2.2 Coupled hydrodynamic-biogeochemical model

A hydrodynamic model based on Princeton Ocean Model (Blumberg and Mellor, 1987) has been developed to simulate the climatological seasonal variation of water circulation in the SIS (Chang et al., 2009; Zhu et al., 2019). The model covered the entire SIS with a range from 32.8°N to 34.8°N and from 130.98°E to 135.5°E (Fig. 1), with two open boundaries south of Bungo Channel and Kii Channel connecting to the Pacific Ocean. The Kanmon Strait, which connects to the Japan Sea, was closed due to the poor water exchange there (Takeoka, 1984). 21 first-class rivers monitored by the Ministry of Land,

Infrastructure and Transportation (MILT) and 45 second-class rivers monitored by the local government were specified along the coastline (Fig. 1b). The fluxes across the air-sea interface, including the momentum flux, mass flux, and heat flux, were from a daily dataset named Grid Point Value of Meso-Scale Model (GPV-MSM) provided by the Japan Meteorological Agency (http://www.jmbsc.or.jp/jp/online/file/f-online10200.html) (Zhu et al., 2019). Four tidal constituents, $M_2$, $S_2$, $O_1$, and $K_1$, were specified at the open boundary of this model (Chang et al., 2009). In addition to the tidal components, the open boundary

conditions also include the subtidal current velocity, temperature, and salinity from a diagnostic model (Guo et al., 2004). The hydrodynamic model provided the current field, temperature field, and diffusion coefficients for the biogeochemical model. For further information on the hydrodynamic model, please refer to Chang et al. (2009) and Zhu et al. (2019).

The biogeochemical model used in this study is a lower-trophic level model, which was employed to depict the annual nutrient and phytoplankton cycle in the coastal seas. It is a nitrogen-based model, and its structure is similar to the one proposed by

Fasham et al. (1990). The model comprises five state variables, namely dissolved inorganic nitrogen (DIN), dissolved inorganic phosphate (DIP), phytoplankton (PHY), zooplankton (ZOO), and particulate organic matter (PON) (Fig. S1). The equations and parametric formulas governing the model were given in the supporting information. The parameter values were obtained from other marine ecosystem models (Fennel et al., 2006; Kishi et al., 2007; Xiu and Chai, 2014), and adjustments were slightly made for the SIS (Table S1).

The initial fields of DIN, DIP, and PHY were obtained from the Broad Comprehensive Water Quality Survey conducted by the Ministry of the Environment, Japan (https://water-pub.env.go.jp/water-pub/mizu-site/mizu/kouiki/dataMap.asp). Specifically, DIN, DIP, and PHY concentrations in January were averaged from 1981 to 2018 and used as the initial fields. As data for PON and ZOO were not available, the initial fields for these state variables were set to zero. This approach was considered reasonable as these variables are known to increase rapidly during the spin-up time (Fennel et al., 2006).

In this study, we examined three primary nutrient sources in the SIS: the land, open ocean, and sediment. We disregarded the DIN loads from the atmospheric deposition in the model. This is because limited studies have been conducted to observe the atmospheric nitrogen deposition flux throughout the SIS, and the reported data covered only a small area and one season (Nakamura et al., 2020). Furthermore, an estimation of the nitrogen budget in SIS reported that the nitrogen loads from

atmospheric deposition were significantly smaller than that from the land through rivers (Yanagi, 1997). As we present later,

because of its small contribution the entire nutrient loads into the SIS, the model without atmospheric nitrogen deposition flux can give reasonable results for the distribution of nutrients and phytoplankton as compared to the observations.

Furthermore, we also excluded nitrogen fixation as a nutrient source for SIS. Hashimoto et al. (2016) reported that nitrogen load from nitrogen fixation was much lower than nitrogen input from rivers in Osaka Bay and Lee et al. (1996) indicated that there was no nitrogen fixation in Hiroshima Bay. In a previous study estimating the nitrogen budget for the SIS, nitrogen

fixation was assumed to be zero (Yamamoto et al., 2008).

Regarding the air-sea interface, we used a daily dataset named third-generation Japanese Ocean Flux Data Sets with the Use of Remote-Sensing Observations (J-OFURO3) to provide the short-wave radiation flux that is used in the photosynthesis (Tomita et al., 2019). The daily dataset was averaged from 2002 to 2013 to obtain a daily climatological value.

The Ministry of the Environment, Japan estimated the total nitrogen (TN) loads of an average value of 471 $mol\ s^{-1}$ from land

to the SIS from 1979 to 2014 based on the unit method (Abo and Yamamoto, 2019; Tomita et al., 2016; Yamamoto, 2003). Yanagi and Ishii (2004) indicated that the TN load estimated by this unit load method did not represent the input of TN into the coastal sea since some parts of the TN load remained on land. Yamamoto et al. (1996) recommended the use of river flow rate to calculate the actual inflow TN load into the SIS and reported that the actual TN load into the SIS calculated using this method was about 48% of that given by the unit load method. Based on this proportion, the average input of TN load to the

SIS was 471×48%=226 $mol\ s^{-1}$.

The proportion of DIN concentration in TN concentration at the first-order rivers of SIS is about 77%, which was estimated by the nutrient data from MILT. Applying this proportion to the TN load from land, the DIN load from land is 226×77%=174 $mol\ s^{-1}$. In our study, we used the daily river discharge and monthly nutrient concentration from MILT averaged over the period from the 1990s to the 2010s to estimate the DIN load from rivers into the SIS. The annual mean of DIN loads from

rivers is 63.85 $mol\ s^{-1}$, which was 37% of the expected DIN load from land into the SIS. To consider the DIN load from land as realistic as possible, the DIN load from rivers was increased to 3 times of the original value to represent the DIN load from land. Consequently, the annual mean of DIN loads from land becomes about 192 $mol\ s^{-1}$. There is a clear spatial variation in DIN loads from land, which shows high DIN loads in the eastern part such as Osaka Bay and Harima Nada (Fig. S2). The seasonal variation in DIN loads, showing high values in July and September and low value in January, is primarily controlled

by the seasonal variation in river discharge. We did not include particle nitrogen input from land.

At the interface between the SIS and the open ocean, it is necessary to specify the nutrient concentrations at two open boundaries. Previous studies have shown that there is a good relationship between nutrient concentration and water temperature in the subsurface water of the Kuroshio Current (Morimoto et al., 2022; Takashi et al., 2006). In this study, we followed this idea and used the water temperature from the hydrodynamic model as a substitute for the observed water temperature to obtain

nutrient concentrations at the open boundaries. To establish the relationship between water temperature and nutrient concentrations, we used the observed monthly water temperature and nutrient concentrations in the Bungo Channel from 1991 to 2005, provided by the Fisheries Research Center, Ehime Prefectural Institute of Agriculture, Forestry and Fisheries, Japan

(Fig. S3). Their strong correlation at the range of low temperatures reflects the inherent nature of water temperature and DIN concentration of the Kuroshio subsurface water. At the range of high temperatures, the DIN concentration is low, which reflects the nutrient-poor Kuroshio surface water. As we lacked observed nutrient concentration data in the Kii Channel, we applied the Bungo Channel relationship to the Kii Channel, given that both channels are influenced by the subsurface water of the Kuroshio Current (Takashi et al., 2006). At the open boundaries in the Bungo Channel and Kii Channel, DIN concentration was high at the bottom throughout the year, with much higher concentrations in summer (Fig. S4). On the other hand, we lacked data for the concentrations of PHY, PON, and ZOO at the open boundaries, so we set their concentrations to zero. While we did not include the input of organic matter from the open ocean into the SIS, we allowed for the export of organic matter from the SIS by using the upwind scheme for tracer at the open boundaries.

At the water-sediment interface, the fluxes of PON ($mmol\ m^{-2}\ day^{-1}$) and DIN ($mg\ m^{-2}\ day^{-1}$) were specified. To determine the downward PON flux to the sediment surface, we followed the method proposed by Ariathurai and Krone (1976) and used Eq. (1):

$$Sediment\ PON\ flux = \begin{cases} C_b w_s \left( \frac{|\tau_b|}{\tau_c} - 1 \right), & |\tau_b| < \tau_c \\ 0, & |\tau_b| \geq \tau_c, \end{cases} \tag{1}$$

where $\tau_c$ is the critical bottom stress. If the magnitude of bottom stress ($\tau_b$) is larger than it, we assume no deposition occurs. We assigned $\tau_c$ a value of 0.14 $N\ m^{-2}$, by considering the sediment distribution in the SIS. $\tau_b$ ($N\ m^{-2}$) is calculated from the hydrodynamic model. $C_b$ is the PON concentration ($mol\ m^{-3}$) in the bottom water. $w_s$ is the sinking velocity of PON which was set at 1 $m\ day^{-1}$.

In addition to the downward PON flux, we also specify an upward DIN flux from the sediment using an empirical formula based on observation data (Tada et al., 2018). For the upward DIN flux from the sediment, we used Eq. (2):

$$Sediment\ DIN\ flux = \begin{cases} 1.802 * exp^{0.1277T} * (TN - 1.301), & TN > 1.301 \\ 0, & TN \leq 1.301, \end{cases} \tag{2}$$

where $T$ is the water temperature (°C) of the bottom water, and $TN$ is the total nitrogen concentration ($mg\ g^{-1}$) at the sediment surface. $T$ was calculated by the hydrodynamic model and applied to the biogeochemical model at each time step. The $TN$ concentration at the sediment surface was the average value from the 1980s to 2010s obtained from the Seto Inland Sea Environmental Information Basic Survey (Ministry of the Environment, Japan; https://www.env.go.jp/water/heisa/heisa_net/setouchiNet/seto/g2/g2cat01/teishitsuodaku/index.html). The annual mean $TN$ concentration was high in Suo Nada, Hiroshima Bay, Harima Nada, and Osaka Bay, but it was below the value of 1.301 $mg\ g^{-1}$ in Bungo Channel, Kii Channel, and Bisan Strait, where no DIN was released from the sediment (Fig. S5). It should be noted that the calculation of DIN flux released from the sediment is somewhat simple in our model. Based on an empirical function obtained from experiments (Tada et al., 2018), we used the annual mean TN concentration and bottom temperature to calculate the sediment DIN flux. This means that we did not consider the instant effect of PON settled to the surface sediment, which can increase the reactive TN concentration in the surface sediment and subsequently higher DIN flux released from the

sediment. In fact, it is difficult to treat such short-term responses of benthic DIN flux to the settled PON (Soetaert et al., 2000) as a source of nutrients because they can be a part of the nutrient cycle within the water column.

The hydrodynamic-biogeochemical model was initiated on the first day of January and stabilized from the third year onwards. Therefore, the simulation results of the third year were used to analyse the seasonal variations of DIN and PHY.

## 2.3 Tracking technique

In our study, we utilized the tracking technique proposed by Kawamiya (2001) and Ménesguen et al. (2006) in conjunction with the hydrodynamic-biogeochemical model. The nutrients in the SIS were considered to have three sources: oceanic, riverine, and benthic nutrients. It needs to note that the nutrients in the sediment are initially from either the land or the open ocean and the sediment acts a temporary storage or a permanent sink of these nutrients. In this study, however, we treat the sediment as the third source to track. This is because the sediment-released nutrients are gaining more attention and are particularly important in shallow waters (Radtke et al., 2019). To represent these sources, we introduced three additional sets of variables, namely $DIN_{ocean}$, $PHY_{ocean}$, $PON_{ocean}$, $ZOO_{ocean}$, $DIN_{river}$, $PHY_{river}$, $PON_{river}$, $ZOO_{river}$, and $DIN_{sediment}$, $PHY_{sediment}$, $PON_{sediment}$, $ZOO_{sediment}$, where the subscripts indicate the source. Each set of variables has its independent equations that are similar to the original equations for the sum of three sources of nutrients, i.e., $DIN$, $PHY$, $PON$, $ZOO$. The complete governing equations for the tracked variables are provided in the Supplement Materials. Regarding the hydrodynamic processes, all tracked variables have the same advection and diffusion terms. However, for the biogeochemical processes, the tracked variables have a different form. Here, we describe how the biogeochemical terms of tracked variables were treated in our study.

Regarding the biogeochemical processes in the model, we calculated the flux that converts one variable to another one (Fig. S1), denoted as $Flux_{A \to B}$, where A was the variable being converted, and B was the target variable. The contribution of each nutrient source to the biogeochemical process depends on its proportion in the total concentration of the variable being converted, as described by Eq. (3), (4), and (5):

$$Flux_{A \to B}^{ocean} = Flux_{A \to B} \frac{[A_{ocean}]}{[A]}, \tag{3}$$

$$Flux_{A \to B}^{river} = Flux_{A \to B} \frac{[A_{river}]}{[A]}, \tag{4}$$

$$Flux_{A \to B}^{sediment} = Flux_{A \to B} \frac{[A_{sediment}]}{[A]}, \tag{5}$$

where the bracket denotes the concentration and $[A] = [A_{ocean}] + [A_{river}] + [A_{sediment}]$. $[A]$ can be any of 4 variables ($DIN$, $PHY$, $PON$, $ZOO$) used in the model.

We initiated the tracking technique from the first day of the fourth year of the hydrodynamic-biogeochemical model, with a value of zero for the initial fields of all the variables originating from the three sources of nutrients. For the open boundary conditions, during the time of inflow, $DIN_{ocean}$ flux had the same values as those used at the open boundaries of the hydrodynamic-biogeochemical model; during the time of outflow, $DIN_{ocean}$ flux was given as the product of $DIN_{ocean}$ and

the outflow velocity. $DIN_{ocean}$ flux was specified to zero at the land-sea interface and the water-sediment interface. $DIN_{river}$ at the land-sea interface was identical to those used in the hydrodynamic-biogeochemical model, but it was set to zero at the water-sediment interface. At the open boundaries, during the time of inflow, $DIN_{river}$ flux was set to zero and during the time of outflow, $DIN_{river}$ flux was given as the product of $DIN_{river}$ and the outflow velocity. $DIN_{sediment}$ was set to have the same flux at the sediment-water interface as that in the hydrodynamic-biogeochemical model, but it was set to have zero flux at the

land-sea interface. At the open boundaries, during the time of inflow, $DIN_{sediment}$ flux was set to zero and during the time of outflow, $DIN_{sediment}$ flux was given as the product of $DIN_{sediment}$ and the outflow velocity. The other model configurations of the tracking technique were the same as those of the hydrodynamic-biogeochemical model. After a spin-up of three years, the annual cycle of each source of nutrients and related particles became stationary.

## 2.4 Validation data

Chang et al. (2009) and Zhu et al. (2019) have conducted a comprehensive comparison between the simulated residual current pattern, monthly water temperature, and monthly salinity and the corresponding observations. We provided a concise summary of their main results in Section 3.1. Additionally, to check whether the mixing ratio between the riverine and oceanic water masses was realistically captured by the model, we compared the simulated sea surface salinity (SSS) with the observations. The observed sea surface salinity was provided by the Fisheries Research Center, Ehime Prefectural Institute of Agriculture,

Forestry and Fisheries, Japan and these data was averaged from 1980 to 2001 to derive the climatological distribution. We obtained the observation data to validate the seasonal and spatial distribution of DIN and PHY from the Broad Comprehensive Water Quality Survey, Ministry of the Environment, Japan (https://water-pub.env.go.jp/water-pub/mizu-site/mizu/kouiki/dataMap.asp). The original data were collected at 121 stations distributed throughout the SIS during January (winter), May (spring), July (summer), and October (autumn) (Fig. 1c). At each station, the data were sampled from two layers,

the upper layer, located 1 m below the sea surface, and the lower layer, positioned 1 m above the sea floor for stations shallower than 50 m. For stations deeper than 50 m, the data for the lower layer were obtained at a fixed depth of 50 m (https://www.env.go.jp/content/900530598.pdf). We used the averaged field from 1981 to 2021, i.e., a climatological field, to validate our model results.

## 2.5 Sensitivity experiments

Understanding the response of SIS to the changes and uncertainty in nutrient inputs from different nutrient sources is crucial for effective nutrient management. To investigate this, we conducted sensitivity experiments by varying the input amount of each nutrient source individually. Nutrient inputs were altered by adding or subtracting the standard deviation of long-term variation of nutrient input based on the climatological input amount. This allowed us to simulate the responses of SIS to the larger and smaller inputs that may occur.

For the DIN load from rivers, we added twice the standard deviation to the climatological value for the upper limit but removed only one standard deviation from the climatological value for the lower limit to avoid negative values. The variation of DIN

loads from rivers was specified in the model by changing the nutrient concentration in river water to avoid changes in the hydrodynamic fields due to the change in river discharge. For DIN load from the open ocean, the nutrient concentration at the open boundaries was added or minus one standard deviation to obtain the upper limit or lower limit, and the open boundary conditions for the hydrodynamic model were not changed. For DIN load from the sediment, TN concentration at the sediment surface in the 1980s was selected as the upper limit and TN concentration in the 2010s was selected as the lower limit because the TN data in the 1980s, 1990s, 2000s, and 2010s show a reduction trend throughout these years. Table 1 summarized the input amount of DIN from the open ocean, rivers, and sediment in each sensitivity experiment.

**Table 1**. Annual mean input amount of DIN ($mol\ s^{-1}$) from the open ocean, rivers, and sediment in sensitivity experiments. "L" means the lower limit case; "U" means the upper limit case. The percentages mean the relative change from the value in the control case.

| Name of cases | From rivers | From open ocean | From sediment | Total input |
|---|---|---|---|---|
| Control | 192 | 799 | 86 | 1077 |
| L-open ocean | 192 | 527 (-34%) | 86 | 805 (-25%) |
| U-open ocean | 192 | 1073 (+34%) | 86 | 1351 (+25%) |
| L-rivers | 130 (-33%) | 799 | 86 | 1015 (-5.8%) |
| U-rivers | 314 (+64%) | 799 | 86 | 1199 (+11%) |
| L-sediment | 192 | 799 | 60 (+30%) | 1051 (-2.4%) |
| U-sediment | 192 | 799 | 136 (+58%) | 1127 (+4.6%) |

## 3 Results

### 3.1 Seasonal and spatial variations of sea surface salinity

Chang et al. (2009) demonstrated that the summer and winter circulation patterns were reproduced by this hydrodynamic model. Significant cyclonic and anticyclonic eddies were developed near the entrance and inner part of Suo Nada, respectively. Moreover, the model also captured the southward current flowing into the western Bungo Channel and the southwestward current in the northern Iyo Nada. Zhu et al. (2019) reported that the warm and saline waters flowed into the SIS through the Bungo Channel and Kii Channel in winter. For the vertical distributions in winter, both the simulated and observed results showed that the water column was well mixed throughout the whole SIS. In summer, both temperature and salinity exhibited a well-mixed pattern around the straits and a well-stratified pattern in the broad basins.

We conducted a comparison between the simulated SSS and the observed one in January (winter), May (spring), July (summer), and October (autumn) (Fig. 2). Throughout the year, the water with a salinity of 33-34 covered the surface layer of Bungo

Channel and Kii Channel, among which the SSS was larger in Bungo Channel than in Kii Channel. In January, the SSS was consistently around 33-34 in most areas of SIS. In May, the SSS decreased in the inshore areas, particularly in the eastern part. In July, the SSS in most inshore areas dropped below 32. In October, the water of lower salinity was refined to the nearshore areas. The hydrodynamic model of this study effectively reproduced the observed SSS distribution in the SIS, indicating a realistic capture of the mixing ratio between riverine and oceanic water masses.

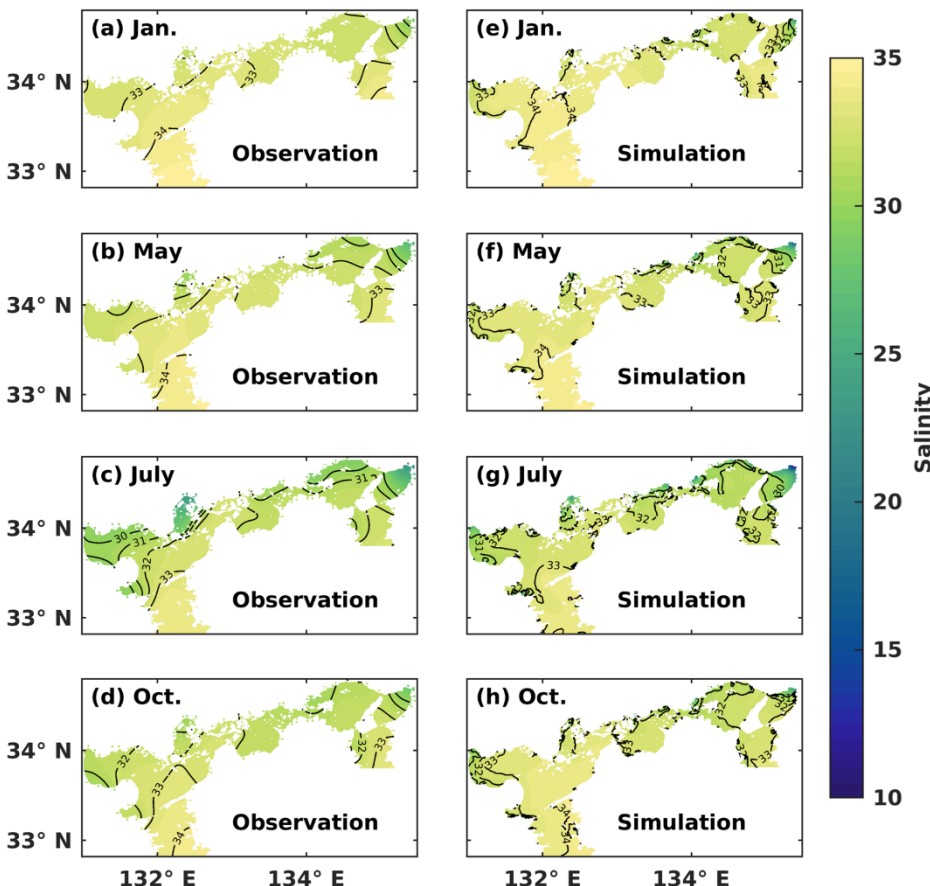

**Figure 2.** Monthly mean **(a-d)** observed and **(e-h)** simulated sea surface salinity.

**3.2 Seasonal and spatial variations of DIN and PHY**

At the upper layer, the observed DIN concentration was low in May (Fig. 3b) and July (Fig. 3c), with a value around 1.0 $mmol\ m^{-3}$, as most of DIN was consumed by PHY for its growth. At the lower layer, the observed DIN concentration

remained relatively stable throughout the year, with a value over 2 $mmol\ m^{-3}$ in most areas, except for May, when it appeared slightly lower at a value around 1.5 $mmol\ m^{-3}$ (Fig. 3f). The observed DIN concentration was well mixed vertically in January

(Fig. 3a, 3e) because the stratification disappeared from late autumn to winter (Chang et al., 2009; Takeoka, 1985). In July, the high DIN concentration at the bottom layers of the Bungo Channel and Kii Channel (Fig. 3g) was due to the intrusion of nutrient-rich open ocean water. Additionally, the observed DIN concentration in July was higher in the eastern part of the SIS, such as Osaka Bay, than in other regions, because the rivers with high DIN loads in the eastern area discharge more DIN into the SIS in summer. The observed DIN concentration was also high at the bottom layer of the central part of Harima Nada in May and July, due to the presence of bottom cold water there. To visualize seasonal patterns, we calculated the mean values of observed and simulated DIN and PHY for the upper and lower years of each subregion. These results are illustrated in Fig. S6. In general, our model reproduced the main seasonal variation and spatial distribution of DIN concentration in the SIS, except for the Bisan Strait and Osaka Bay, where the simulation results were lower than the observations. This was probably because the DIN loads from sewage sources were not considered in the model. The comparison of simulated DIN concentration versus observations suggests that the model works well in the areas with low concentrations ($<10$ $mmol\ m^{-3}$) but underestimates the high concentrations in some nearshore areas. Their linear correlation coefficient was 0.82 (Fig. 3i).

At the upper layer, the observed PHY concentration was high in May and October, with values over 1 $mg\ Chla\ m^{-3}$ (Fig. 4b, 4d). At the lower layer, the observed PHY concentration remained at a low value below 0.5 $mg\ Chla\ m^{-3}$ all year round in most areas (Fig. 4e-4h). However, it was high in the nearshore areas, particularly around the estuaries. In January, the observed PHY concentration was similar between the upper and lower layers (Fig. 4a, 4e). The difference in PHY concentration between the upper and lower layers was larger in the simulation than in the observation. The simulated PHY concentration was lower than the observations in the nearshore areas. The mean values of observed and simulated DIN and PHY for the upper and lower years of each subregion are illustrated in Fig. S7. The comparison of simulated PHY concentration versus observations suggests that the model works well in areas with low concentrations ($<5$ $mg\ Chla\ m^{-3}$) but underestimates the high concentrations in the nearshore areas. Their linear correlation coefficient was 0.63 (Fig. 4i).

It should be noted that the underestimations for DIN and PHY occur only in the nearshore areas (surface salinity <30.6 in July refer to Ministry of the Environment, Japan), whose volume shares 5% of the whole SIS. In other words, the model can reproduce the seasonal variations and the spatial distribution of DIN and PHY in the other 95% volume of the SIS, which is sufficient for our purpose to examine the impacts of oceanic nutrients on the inventories of DIN and PHY in the SIS.

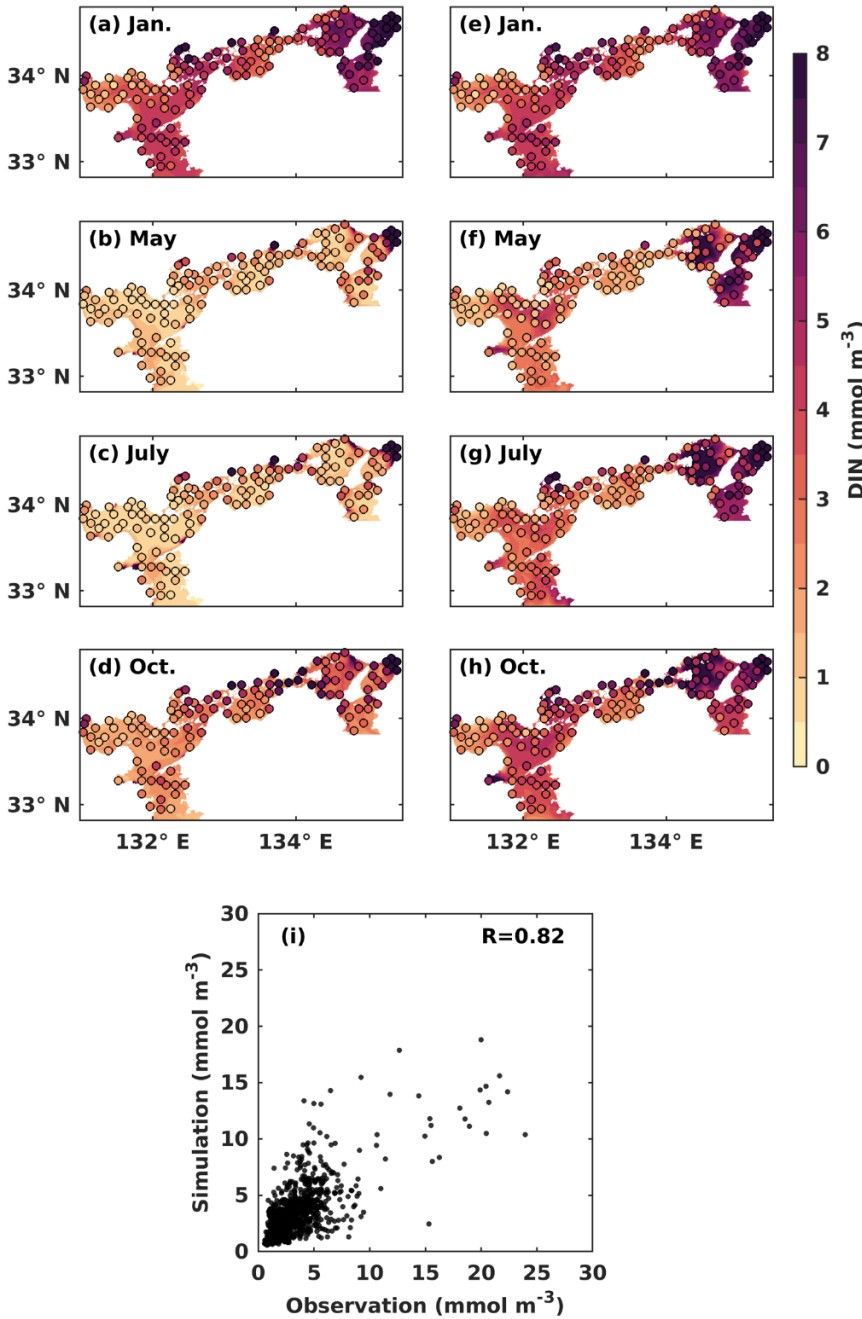

**Figure 3**. Monthly mean DIN concentration ($mmol\ m^{-3}$) at the **(a-d)** upper and the **(e-h)** lower layers. Solid circles represent the observed values, and the background colour represents the simulation values. **(i)** Comparison between observed DIN concentrations versus simulated DIN concentrations.

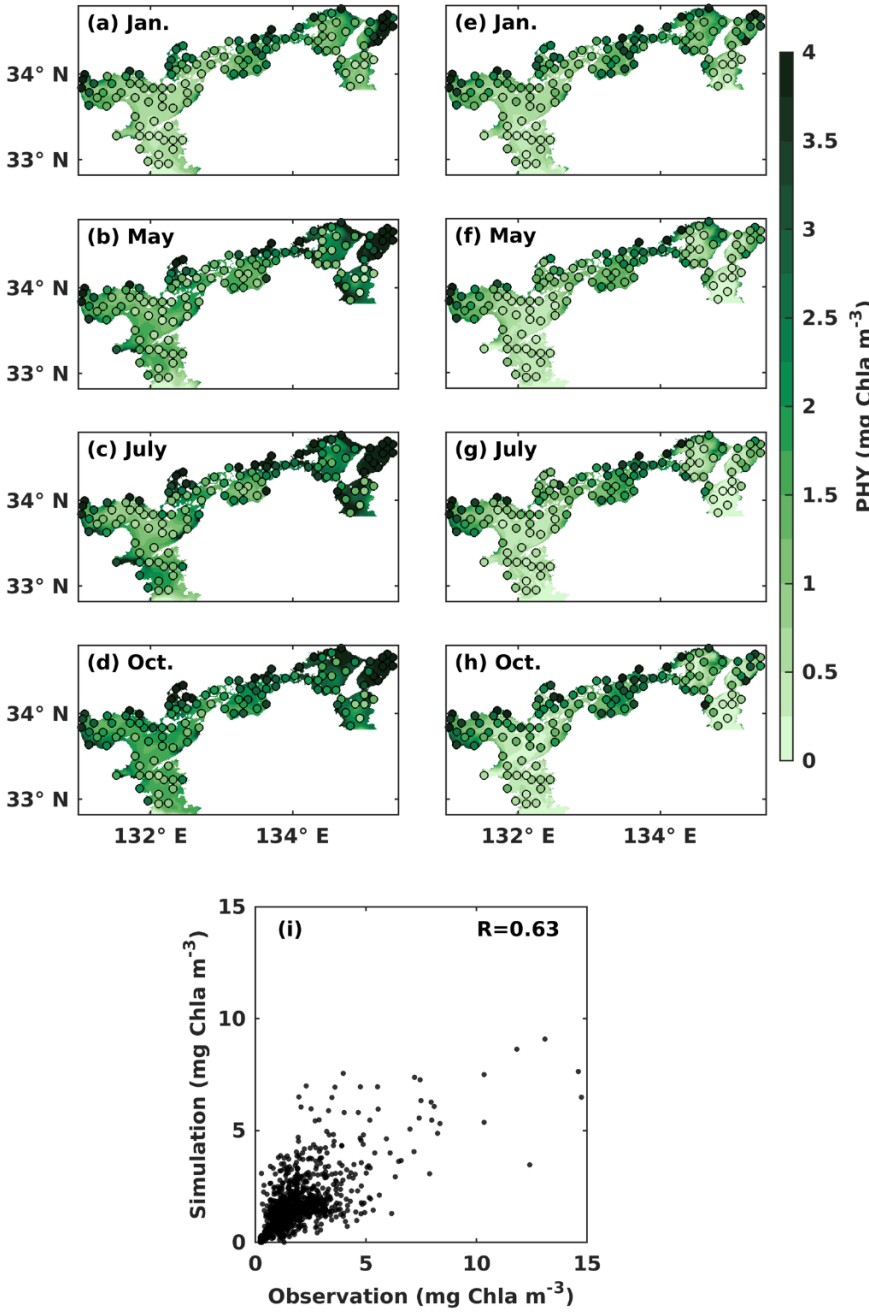

**Figure 4**. Monthly mean PHY concentration ($mg\ Chla\ m^{-3}$) at the **(a-d)** upper and the **(e-h)** lower layers. Solid circles represent the observed values, and the background colour represents the simulation values. **(i)** Comparison between observed PHY concentrations versus simulated PHY concentration.

### 3.3 Distribution of oceanic, riverine, and benthic DIN in the first year of simulations

The materials related to the DIN originating from the open ocean, rivers, and sediment were all set to zero at the initial stage of the simulation. After DIN loads were transported into the SIS, DIN concentration from each nutrient source gradually increased in the SIS until reaching the steady state of the annual cycle. The water-column averaged DIN concentration in the first year of tracking simulation was used to exhibit the pathway of DIN from each nutrient source into the SIS (Fig. 5).

The oceanic DIN intruded into the SIS from the Bungo Channel and Kii Channel and then entered the Iyo Nada and Osaka Bay. Gradually, it arrived the Suo Nada, Aki Nada, and Harima Nada. In the western part of the SIS, the oceanic DIN continued to intrude into Hiroshima Bay and Hiuchi Nada. On the other hand, in the eastern part of the SIS, the oceanic DIN was mainly concentrated at Kii Channel, Osaka Bay, and Harima Nada, but its concentration increased (Fig. 5a-5f). The oceanic DIN concentration at Bisan Strait started to increase at the end of the first year. On the other hand, the riverine DIN intruded into the SIS from the estuaries, especially in Osaka Bay and Harima Nada (Fig. 5g-5l). The benthic DIN concentration increased from the nearshore areas of Suo Nada, Hiroshima Bay, Hiuchi Nada, Harima Nada, and Osaka Bay (Fig. 5m-5p).

After one year of simulation, DIN concentrations from the open ocean, rivers, and sediment have already occupied most areas of the SIS (Fig. S8). We used the results in the third year of the tracking simulation, which has reached the steady state of the annual cycle, for the following sections.

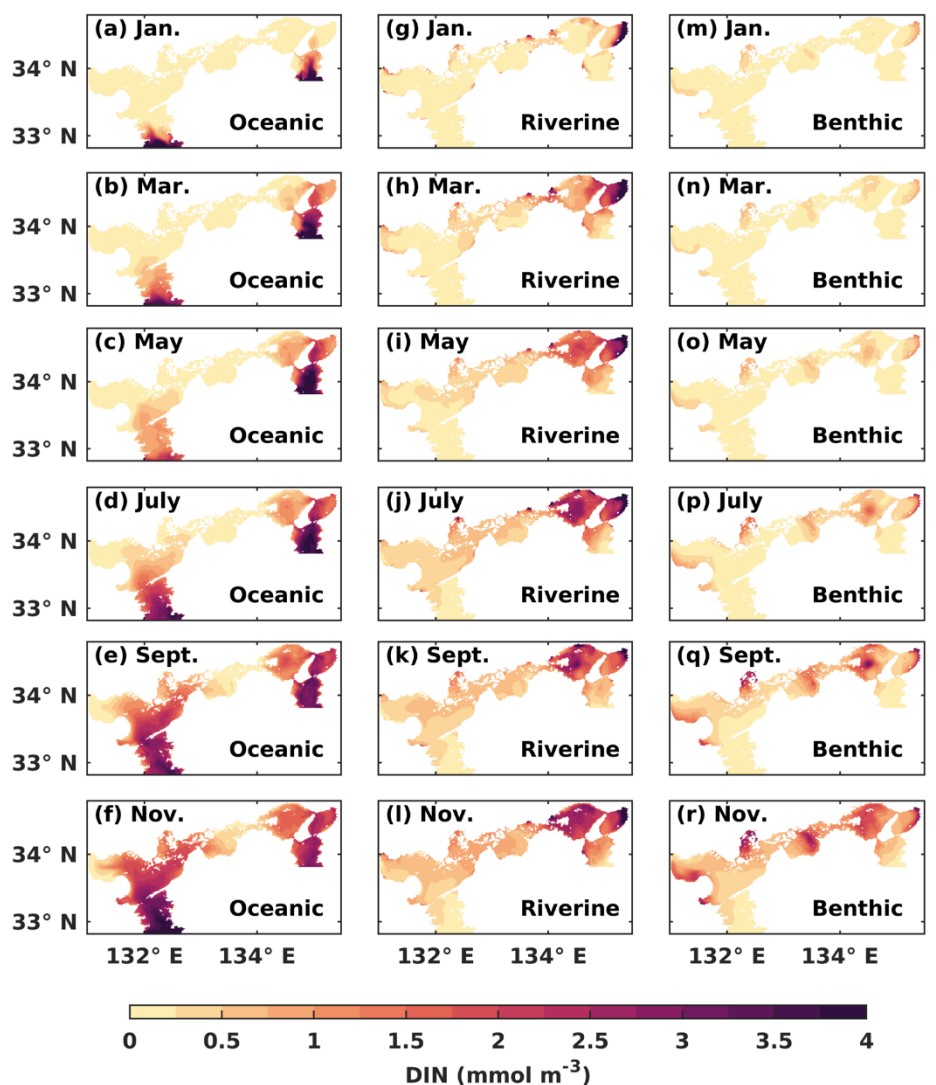

**Figure 5**. The water column averaged DIN concentration ($mmol\ m^{-3}$) from the open ocean, rivers, and sediment in the first year of the tracking simulation.

### 3.4 Contribution of different nutrient sources to DIN and PHY

The oceanic DIN concentration in the entire SIS showed a significant seasonal variation, with a decrease from 2.7 $mmol\ m^{-3}$ in January to 1.7 $mmol\ m^{-3}$ in June, followed by an increase until December, similar to the total DIN concentration (Fig. 6h). The proportion of oceanic DIN to the total DIN did not show an obvious seasonal variation, ranging between 57% and 66%, with an annual mean of 61% (Fig. 6i). The riverine and benthic DIN concentrations were much smaller than oceanic DIN

concentration and remained around 0.89 $mmol\ m^{-3}$ and 0.52 $mmol\ m^{-3}$, respectively. The annual average contribution of riverine and benthic sources to DIN concentration was 25% and 14%, respectively.

Regarding the annual mean spatial variation, the oceanic DIN concentration averaged over the entire depth was higher than 2.5 $mmol\ m^{-3}$ in the Bungo Channel and Kii Channel, while it was lower than 0.5 $mmol\ m^{-3}$ in Bisan Strait and Bingo Nada (Fig. 6b). The oceanic contribution to DIN concentration decreased from the southern boundary areas to the inner part of the SIS and exhibited lower values in the nearshore areas. For different sub-regions, the proportion of oceanic nutrients was high at Bungo Channel (83%), Kii Channel (68%), and Iyo Nada (65%) and low at Bisan Strait (17%), Bingo Nada (22%) (Fig. 8a).

The riverine concentration was highest in Harima Nada and Osaka Bay, due to large terrestrial input, and the corresponding contribution was also highest in these two sub-regions with a value of about 50%. The benthic concentration was higher in Bingo Nada, Harima Nada, and Hiroshima Bay, while its contribution was higher in Bingo Nada, with a value of about 46%. The volume averaged PHY concentration supported by oceanic DIN also exhibited a significant seasonal variation, with an obvious spring bloom in March (0.61 $mg\ Chla\ m^{-3}$) (Fig. 7h). However, the seasonal variation in the oceanic contribution to

the total PHY concentration remained close to the annual mean value of 46% (Fig. 7i). The PHY concentrations supported by riverine DIN and benthic DIN also change much with seasons, with an annual average of 0.38 $mg\ Chla\ m^{-3}$ and 0.20 $mg\ Chla\ m^{-3}$, respectively. The annual average contribution of riverine and benthic sources to PHY concentration was 35% and 19%, respectively.

For the spatial variation of annual mean, the oceanic PHY concentration averaged over the entire depth remained at 0.5

$mg\ Chla\ m^{-3}$ in most areas of the SIS (Fig. 7b). However, the oceanic contribution to PHY concentration exhibited an obvious spatial pattern, decreasing from the southern boundary areas into the inner part of the SIS and showing lower values in the nearshore areas, similar to DIN. For different sub-regions, the oceanic contribution was much higher at Bungo Channel (78%), Kii Channel (46%), and Iyo Nada (59%) and lower at Bisan Strait (15%), Bingo Nada (21%) (Fig. 8b). The riverine contribution was higher at Harima Nada (56%) and Osaka Bay (62%), while the benthic contribution was higher at Bingo

Nada (42%) and Bisan Strait (32%).

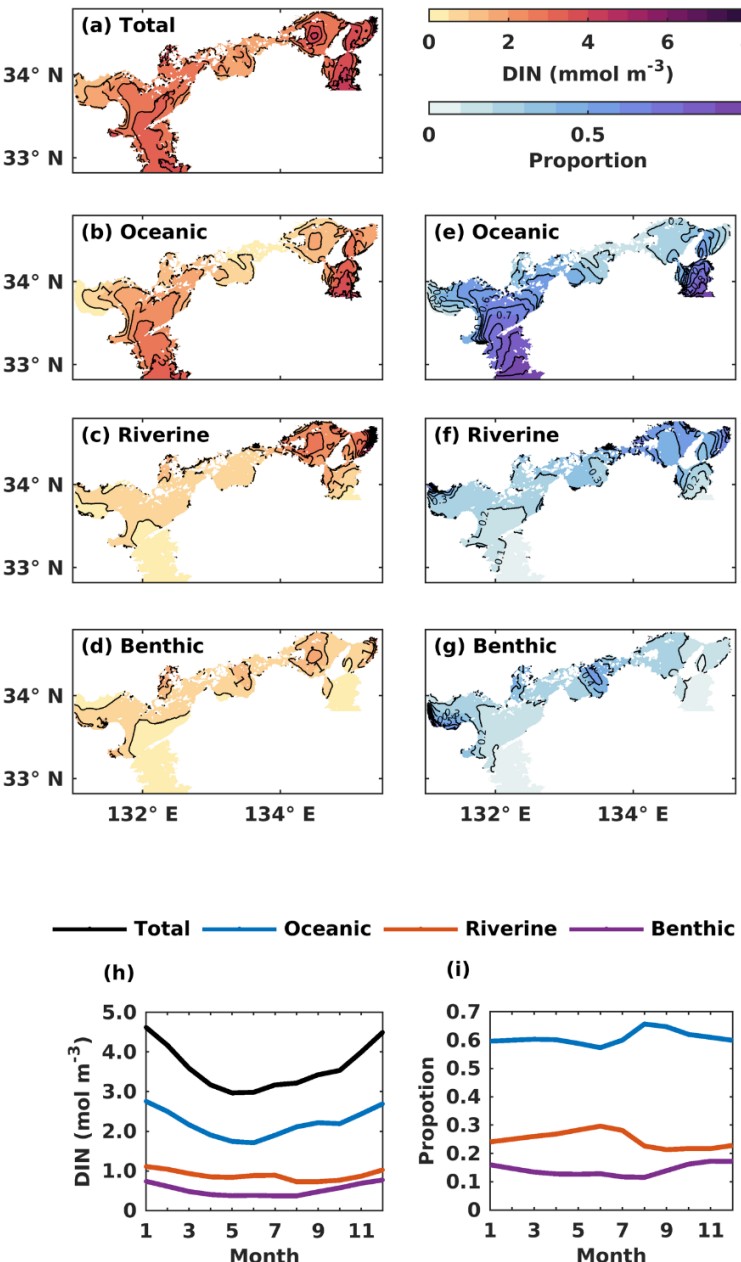

**Figure 6. (a-d)** The annual mean of water column-averaged DIN concentration ($mmol\ m^{-3}$) originating from the open ocean, rivers, and sediment. **(e-g)** Contributions of the open ocean, rivers, and sediment to total DIN concentration. **(h)** The monthly average oceanic, riverine, and benthic DIN concentration ($mmol\ m^{-3}$) in the whole SIS. **(i)** The monthly variations of oceanic, riverine, and benthic contributions to total DIN concentration in the whole SIS.

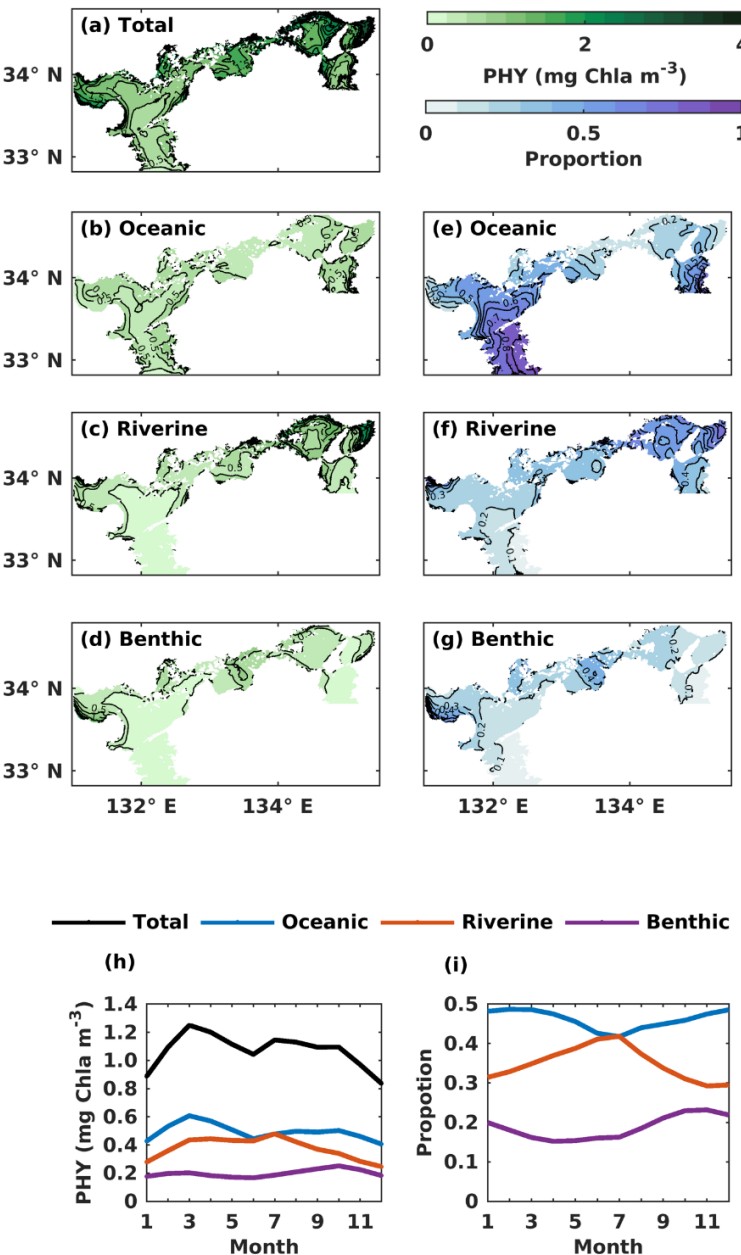

**Figure 7. (a-d)** The annual mean of water column-averaged PHY concentration ($mg\ Chla\ m^{-3}$) originating from the open ocean, rivers, and sediment. **(e-g)** Contributions of the open ocean, rivers, and sediment to total PHY concentration. **(h)** The monthly average oceanic, riverine, and benthic PHY concentration ($mg\ Chla\ m^{-3}$) in the whole SIS. **(i)** The monthly variations of oceanic, riverine, and benthic contributions to total PHY concentration in the whole SIS.

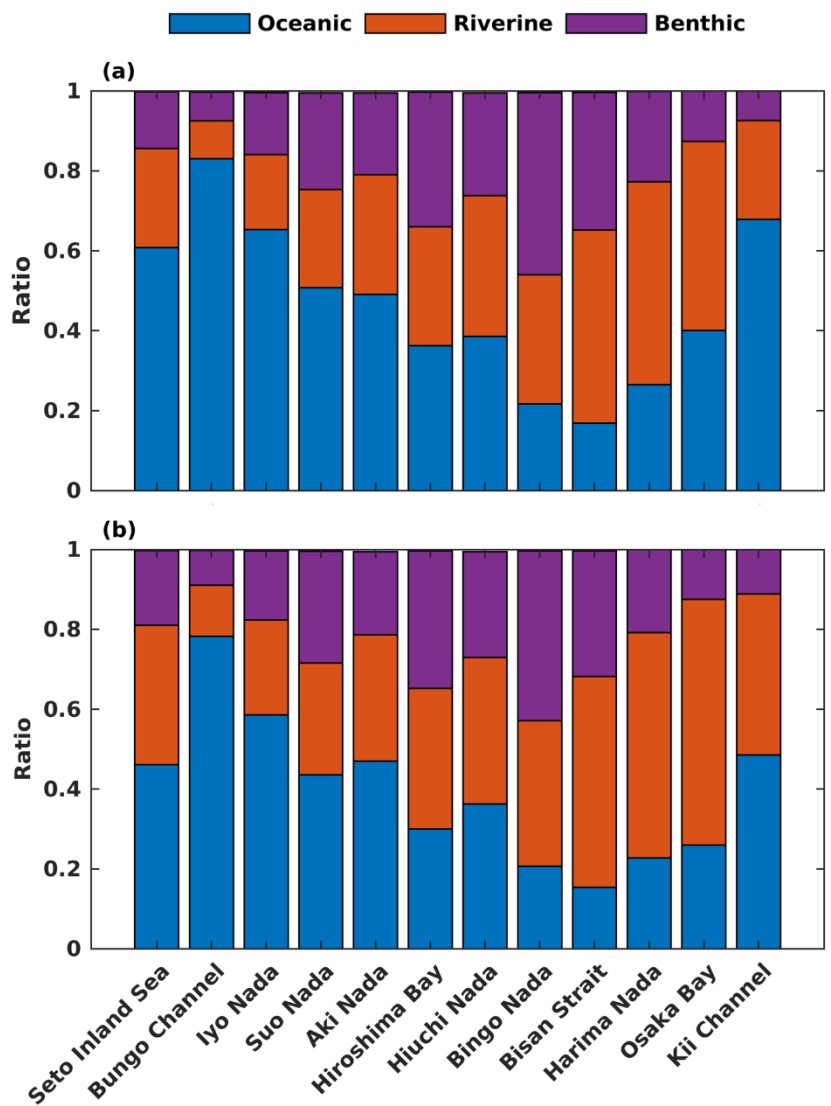

**Figure 8**. Contributions of the open ocean, rivers, and sediment to total **(a)** DIN and **(b)** PHY concentrations in sub-regions of SIS.

### 3.5 DIN transport across the boundary between SIS and open ocean

The oceanic DIN flux was primarily transported from the open ocean to the SIS at the lower layers, reflecting the intrusion of nutrient-rich Kuroshio subsurface water from the bottom, and out of the SIS at the upper layers and eastern part of the open

boundaries (Fig. S9). In terms of the seasonal variation of DIN transport at the Bungo Channel, the onshore transport of oceanic DIN was highest in January and July, with a value of 707 $mol\ s^{-1}$ and 664 $mol\ s^{-1}$ respectively, and lowest in April and May with a value of 236 $mol\ s^{-1}$ and 226 $mol\ s^{-1}$, respectively (Fig. 9a). There is also offshore transport of oceanic DIN across

the open boundary. Its value was highest in January and February, with a value of -648 $mol\ s^{-1}$ and -550 $mol\ s^{-1}$, respectively, and lowest in April and May, with a value of -244 $mol\ s^{-1}$ and -215 $mol\ s^{-1}$, respectively. Consequently, the net oceanic DIN transport was highest in July and August, with a value of 275 $mol\ s^{-1}$ and 215 $mol\ s^{-1}$, respectively, and lowest in April and May, with a value of -8 $mol\ s^{-1}$ and 10 $mol\ s^{-1}$, respectively (Fig. 9a).

The seasonal variation of oceanic DIN transport across the Kii Channel was somewhat different from that of the Bungo Channel.

The onshore transport of oceanic DIN was highest in July and August, with a value of 407 $mol\ s^{-1}$ and 483 $mol\ s^{-1}$, respectively, and lowest in April and May, with a value of 199 $mol\ s^{-1}$ and 215 $mol\ s^{-1}$, respectively. The offshore transport of oceanic DIN was highest in July and August, with a value of -277 $mol\ s^{-1}$ and -329 $mol\ s^{-1}$, respectively, and lowest in March and April, with a value of -123 $mol\ s^{-1}$ and -112 $mol\ s^{-1}$, respectively. Consequently, the net oceanic DIN transport was highest in July and August, with a value of 130 $mol\ s^{-1}$ and 154 $mol\ s^{-1}$, respectively, and lowest in May and November,

with a value of 70 $mol\ s^{-1}$ and 67 $mol\ s^{-1}$, respectively (Fig. 9b).

The onshore transport of oceanic DIN has an annual mean of 507 $mol\ s^{-1}$ at Bungo Channel and 292 $mol\ s^{-1}$ at Kii Channel, whose sum is 799 $mol\ s^{-1}$. The offshore transport of oceanic DIN has an annual mean of -397 $mol\ s^{-1}$ at Bungo Channel and of -199 $mol\ s^{-1}$ at Kii Channel, whose sum is -596 $mol\ s^{-1}$. The values produce an annual mean of net oceanic DIN transport of 110 $mol\ s^{-1}$ at Bungo Channel and 93 $mol\ s^{-1}$ at Kii Channel, whose sum is 203 $mol\ s^{-1}$.

At the south of Bungo Channel, Morimoto et al. (2022) reported that a net of 385 $mol\ s^{-1}$ oceanic DIN was transported from the open ocean to the SIS in July and August based on simulated water volume and DIN concentration derived from water temperature. It was 245 $mol\ s^{-1}$ in our study. The reason that our estimate is less than theirs may be caused by the outward DIN transport. We used the DIN concentration calculated by the hydrodynamic-biogeochemical model, which was larger than their DIN concentration. Another possible reason is that the simulation of July and August 2018 by Morimoto et al. (2022) is

for an extremely heavy rain case, which induced a stronger outflow in the surface layer and a stronger inflow in the bottom layer than those in our simulation. Fujiwara et al. (1997) reported about 168 $mol\ s^{-1}$ of DIN through Kii Channel from the open ocean in August of 1985. Kasai et al. (2001) estimated the net DIN transport through Kii Channel into SIS in August of 1996 was 111 $mol\ s^{-1}$ which was similar to that of Fujiwara et al. (1997). We estimated about 150 $mol\ s^{-1}$ of DIN was from the open ocean to the SIS through Kii Channel in August.

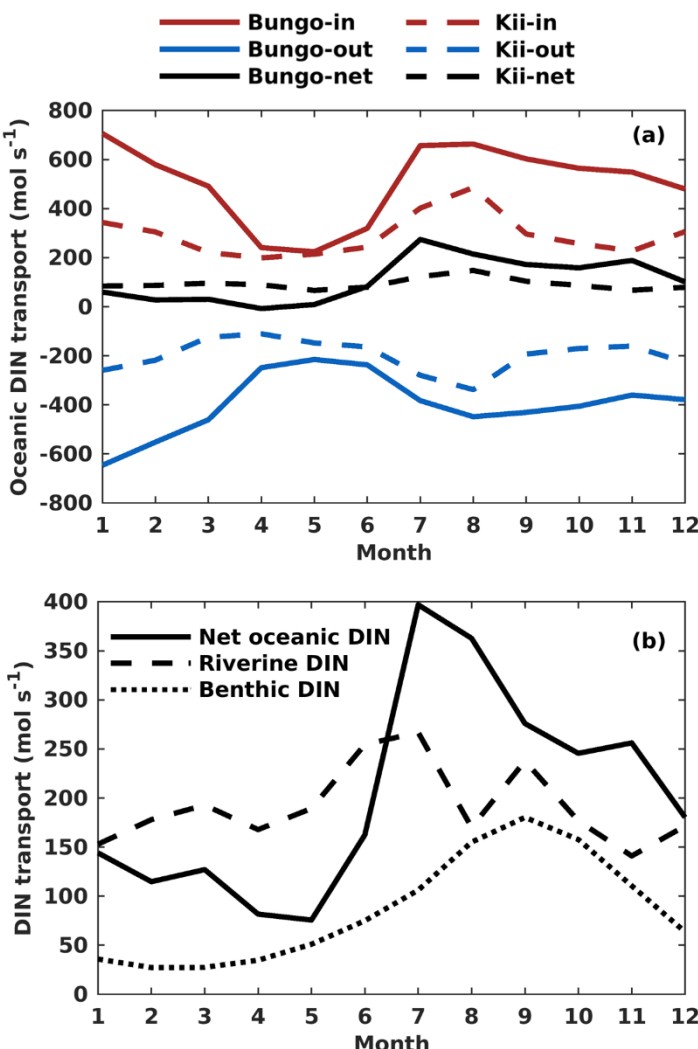

**Figure 9.** **(a)** Monthly variation of onshore, offshore, and net transport of ocean DIN ($mol\ s^{-1}$) across the Bungo Channel and Kii Channel. **(b)** Monthly variation of DIN loads ($mol\ s^{-1}$) from the open ocean, rivers, and sediment.

## 4 Discussion

### 4.1 Role of open ocean in DIN inventory of different shelf seas

Our simulation reveals that the annual mean inventory of DIN in the whole SIS was $31.7 \times 10^8\ mol$, among which that originating from the open ocean, rivers, and sediment was $19.2 \times 10^8$, $7.89 \times 10^8$, and $4.62 \times 10^8\ mol$, respectively, sharing a proportion of 61%, 25%, and 14%, respectively (Fig. 10). The oceanic contribution to DIN inventory in our simulation (61%)

was less than 81% proposed by (Yanagi and Ishii, 2004) who did not consider biogeochemical processes and nutrient release from sediment.

In terms of PHY, the annual mean inventory supported by DIN from the open ocean, rivers, and sediment was $2.75 \times 10^8$, $2.10 \times 10^8$, and $1.11 \times 10^8$ $mol\ N$, respectively, which imply a proportion of 46%, 35%, and 19%, respectively, in the whole inventory (Fig. 10). Although being smaller than the contribution to DIN, the open ocean is still a dominant contributor to PHY in the SIS. If we consider the total nitrogen, which is defined as the sum of DIN, PHY, ZOO, and PON in this study, the open ocean contributes 53% to its inventory in the SIS.

In the ECS, a marginal sea in the Northwest Pacific Ocean with a broad continental shelf, the open ocean plays a dominant role in areas shallower than 200 m, contributing 57% of the DIN inventory (Zhang et al., 2019). Even though the ECS is more open to the open ocean, our simulation showed that the open ocean contributed more to the DIN inventory in the SIS compared to the ECS. In a shallow estuary located on the east coast of Jutland, Denmark, which is well-ventilated with the open ocean, the open ocean contributed less than 40% to the DIN inventory in most areas of the estuary (Timmermann et al., 2010). In the Northern Gulf of Mexico, another marginal sea receiving large amounts of nutrients from rivers, the open ocean contributed less than 15% of the total nitrogen inventory in the shallow regions, where rivers contribute more than 85% (Große et al., 2019). Our study also found that the open ocean contributed less than 30% in the nearshore areas with shallow water depth and large riverine nutrient inputs (Fig. 6e, 7e). Apparently, the oceanic contribution to the nutrient inventory in these water bodies varies largely and should strongly depend on its cycling rate in the low-trophic system of the shelf sea.

## 4.2 DIN and PHY material flows

To understand the different proportions of oceanic, riverine, and benthic nutrients in the inventory of DIN in the SIS, we present its related material flows here (Fig. 10).

The annual mean budget flows of DIN in the SIS can be described using Eq. (6), where the amount of DIN delivered to the SIS from external sources is equal to the sum of the amount consumed by biogeochemical processes that transfer DIN to the biological particles and the amount exported to the open ocean through open boundaries in the form of DIN.

$$Input\ transport = Output\ amount + Net\ biogeochemical\ processes\ of\ DIN, \qquad (6)$$

$$Net\ biogeochemical\ processes\ of\ DIN =$$
$$-Photosynthesis + Respiration + Mineralization + Excretion, \qquad (7)$$

Our study found that the input amount of DIN from the open ocean, rivers, and sediment was 799, 192, and 86 $mol\ s^{-1}$, respectively, with a ratio of 74%, 18%, and 8%. The output amount of oceanic, riverine, and benthic DIN to the open ocean was -596, -42, and -17 $mol\ s^{-1}$, respectively, with a ratio of 91%, 6%, and 3%. The net biogeochemical process of DIN from the open ocean, rivers, and sediment was -202, -150, and -68 $mol\ s^{-1}$, respectively, with a ratio of 48%, 36%, and 16%. These results indicate that only 25% of the input oceanic DIN was involved in biogeochemical processes, and 75% was returned to the open ocean (Fig. 10b). Therefore, the role of oceanic nutrients may be overestimated in terms of the input amount. In

contrast, riverine and benthic DIN inputs transport was smaller than oceanic DIN, but 78% and 79% of their inputs were involved in biogeochemical processes, indicating more bioactive effects (Fig. 10c, 10d). In the North Sea, which is a semi-enclosed sea connected to North Atlantic Ocean through its northern boundary, the open ocean contributed 75% of the total nitrogen budget in the north open area (Vermaat et al., 2008). Similarly, in the ECS, the open ocean contributed 72% of the DIN input amount in areas shallower than 200 m (Zhang et al., 2019). As shown by Zhang et al. (2019), the contribution of

oceanic DIN to primary production in the ECS is lower than the proportions of DIN input (72%). Such a difference suggests a different efficiency of DIN with different origins.

    In our simulation, we did not allow external sources to bring PHY into the SIS through open boundaries. Therefore, the PHY generated through biogeochemical processes inside the SIS was transported to the open ocean through the open boundaries. The equation governing the budget of PHY is represented by Eq. (8). The net biogeochemical processes of PHY are defined

in Eq. (9)

$$0 = Output\ transport + Net\ biogeochemical\ processes\ of\ PHY, \tag{8}$$

$$Net\ biogeochemical\ processes\ of\ PHY =$$
$$Photosynthesis - Respiration - Grazing - Mortality, \tag{9}$$

    The oceanic proportion of photosynthesis was 41%, which is lower than the proportion of oceanic DIN in the total inventory.

This may be due to the fact that the oceanic DIN was primarily distributed in the lower layer of areas with deep depth in the SIS (Fig. 6b). As a result, some of the oceanic DIN in the deep layer may not be available for photosynthesis due to light limitation, leading to a lower proportion of oceanic photosynthesis compared to oceanic DIN. In contrast, riverine and benthic DIN were mainly distributed in the areas with shallow depths where the light condition is good (Fig. 6c, 6d).

    The net biogeochemical process of PHY defined by Eq. (9) was 64, 20, and 6 $mol\ s^{-1}$ for the oceanic riverine, and benthic

DIN, respectively, which are exactly the same as the output amount of PHY from the open boundaries. Their ratio of 71%, 22%, and 7% suggests that the oceanic PHY was more likely to leave the SIS.

    The inventory of ZOO and PON depends on that of PHY and consequently the nutrients from the open ocean support more than half of ZOO and PON in the SIS. The export of biological particles from SIS is important for a material balance in the sea. In the SIS, the horizontal export flux of biological particles (PHY+ZOO+PON) to the open ocean is 229 $mol\ s^{-1}$ (Fig.

10a) and the vertical export flux of biological particles to the sediment is 190 $mol\ s^{-1}$, whose ratio is 229 $mol\ s^{-1}$/190 $mol\ s^{-1} \approx 1.2$:1. If we examine them for the different origins of nutrients, this ratio changes. For oceanic nutrients (Fig. 10b), the horizontal export of biological particles has a flux of 140 $mol\ s^{-1}$ while the vertical export has a value of 61 $mol\ s^{-1}$, whose ratio is 140 $mol\ s^{-1}$/61 $mol\ s^{-1} \approx 2.3$:1; for the riverine nutrients (Fig. 10c), the horizontal export has a flux of 68 $mol\ s^{-1}$ while the vertical export has a value of 82 $mol\ s^{-1}$, whose ratio is 68 $mol\ s^{-1}$/82 $mol\ s^{-1} \approx 0.83$:1; for the benthic

nutrients (Fig. 10d), the horizontal export has a flux of 21 $mol\ s^{-1}$ while the vertical export has a value of 47 $mol\ s^{-1}$, whose ratio is 21 $mol\ s^{-1}$/47 $mol\ s^{-1} \approx 0.45$:1. Considering the spatial distribution of these different origins of nutrients, the difference in ratios is likely a natural result of water exchange between the SIS and the open ocean.

If the proportion of PON flux settled on the sediment surface produced by the oceanic and riverine nutrients were taken as the proportion of DIN flux released from the sediment, among the 86 $mol\ s^{-1}$ of DIN from the sediment, 37 $mol\ s^{-1}$ has an origin

in the open ocean and 49 $mol\ s^{-1}$ from rivers.

There have been some studies that estimated the TN budget in the SIS based on certain assumptions. Yanagi (1997) reported that 392 $mol\ s^{-1}$ of TN was transported from the land, and 31 $mol\ s^{-1}$ from atmospheric deposition. Among these inputs, 358 $mol\ s^{-1}$ of TN was transported to the open ocean at the open boundaries, while 64 $mol\ s^{-1}$ was buried in the sediment. Our study revealed that 192 $mol\ s^{-1}$ of DIN was from rivers. After undergoing biogeochemical processes, 110 $mol\ s^{-1}$ was

transported to the open ocean as sum of DIN, PHY, PON, and ZOO, while 82 $mol\ s^{-1}$ was buried in the sediment in the form of PON. It is important to consider a few factors when comparing our results to those in Yanagi (1997). First, we did not introduce the other types of nitrogen from the land. Additionally, the TN load used in Yanagi (1994) was obtained in 1982, a period characterized by severe terrestrial pollution. As a result, the nitrogen load from land in our study was lower compared to those reported by Yanagi (1997).

Fujiwara et al. (2006) also estimated the TN budget in the SIS and clarified the land origin and open ocean origin, showing that 330 $mol\ s^{-1}$ of TN was supplied from the land to SIS, of which 297 $mol\ s^{-1}$ was transported to the open ocean and 33 $mol\ s^{-1}$ was buried in the sediment. Fujiwara et al. (2006) also reported that the net input of TN from the open ocean was 50 $mol\ s^{-1}$, which was buried in the sediment. In our study, the TN originating from the open ocean has a net input of 61 $mol\ s^{-1}$, all of which were buried in the sediment.


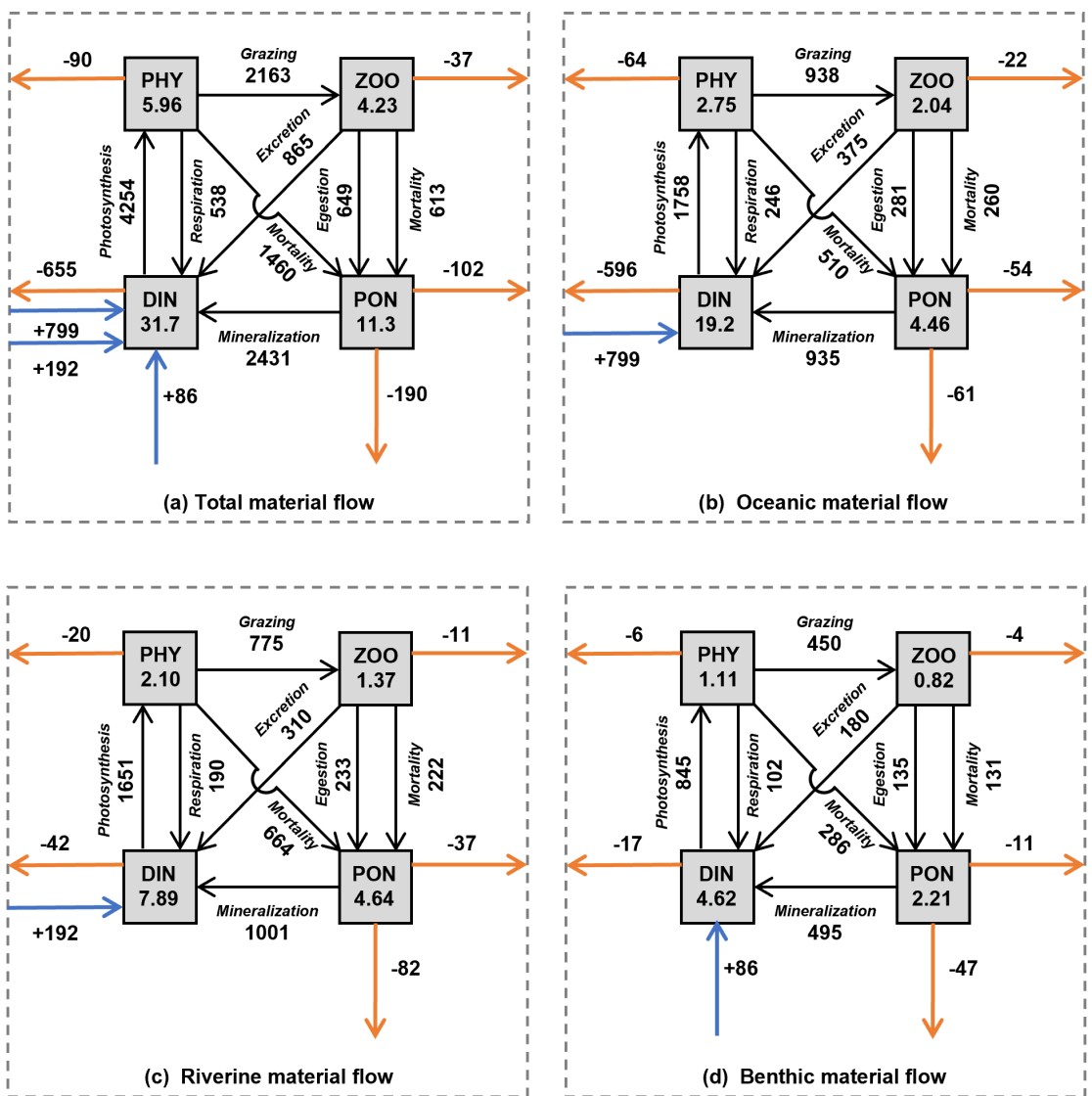

**Figure 10**. The annual mean of material inventories and material flows in the whole SIS. **(a)** Total inventories and material flows **(b)** Oceanic inventories and material flows **(c)** Riverine inventories and material flows **(d)** Benthic inventories and material flows. The values in the rectangles represent the inventory ($\times 10^8\ mol$) of each material in the SIS. The values next to the black solid lines with an arrow represent the flux of biogeochemical processes ($mol\ s^{-1}$). The values next to the horizontal orange solid lines with an arrow represent the horizontal flux from the SIS to the open ocean ($mol\ s^{-1}$). The values next to the vertical orange solid lines with an arrow represent the vertical flux to the sediment ($mol\ s^{-1}$). The values next to the blue solid line with a vertical arrow represent the flux of benthic DIN into the SIS.

### 4.3 Response of the SIS to changes in nutrient input

When the DIN load from the open ocean was reduced by 33% (case of L-open ocean in Table 1), there was a corresponding decrease in the concentration of oceanic DIN by 34% ((1.44-2.19)/2.19), as well as a decrease in the concentration of oceanic PHY by 28% ((0.35-0.49)/0.49) (Table 2). As a result, the contribution of oceanic DIN and PHY to the total concentration decreased by 10%, while those of riverine and benthic DIN and PHY increased with a range between 3% to 6%. Conversely, when the DIN load from the open ocean was increased by 34% (case of U-open ocean in Table 1), the concentration of oceanic

DIN increased by 35%, and that of oceanic PHY by 24% (Table 2). Then, the contribution of oceanic DIN and PHY to the total concentration increased by 6%, while that of riverine and benthic DIN and PHY decreased by 2% to 4%.

Similarly, when the DIN load from rivers was reduced by 33% (case of L-rivers), the concentration of riverine DIN and PHY decreased by 30% ((0.62-0.89)/0.89) and 26% ((0.28-0.38)/0.38), respectively, and their contribution to the total concentration decreased by 6% and 7%, respectively (Table 2). On the other hand, when the DIN load from rivers was increased by 64%

(case of U-rivers), the concentration of riverine DIN and PHY increased by 72% ((1.53-0.89)/0.89) and 42% (0.54-0.38)/0.38, respectively, and their contribution to the total concentration increased by 11% and 10%, respectively. In these two cases, the contribution of oceanic and benthic DIN and PHY changed with a range of less than 9%.

When the DIN load from the sediment was reduced by 30% (case of L-sediment), the concentration of benthic DIN and PHY decreased by about 20% ((0.41-0.52)/0.52 and (0.16-0.20)/0.20), and their contribution to the total concentration decreased by

about 4%. When the DIN load from the sediment was increased by 58% (case of U-sediment), the concentration of benthic DIN and PHY increased by 29% ((0.67-0.52)/0.52) and 25% ((0.25-0.20)/0.20), respectively, and their contribution to the total concentration increased about 4%.

Overall, the change in the DIN load from a specific nutrient source led to the same order change in the corresponding DIN and PHY concentrations, but a small range of change in the contribution of all nutrient sources. Specifically, despite significant

changes in DIN loads from rivers and sediment, their contribution to the SIS was only minimally impacted due to their low proportion. This can be understood by the relative change in the inventory of total DIN and total PHY in Table 2. In these experiments, the changes in oceanic nutrients can cause more than a 20% change in total DIN inventory and about a 10% change in PHY inventory. The changes in riverine nutrients can cause less than a 20% change in total DIN inventory and about a 10% change in PHY inventory. However, the changes in benthic nutrients can cause only ~4% changes in the total DIN

inventory and PHY inventory.

**Table 2**. Annual mean oceanic, riverine, and benthic DIN concentration ($mmol\ m^{-3}$) and PHY concentration ($mg\ Chla\ m^{-3}$) and their proportion in total DIN and PHY concentration in the entire SIS in the sensitivity experiments. The percentage in column 6 is the relative change to the values in the case of "Control".

| Name | | Oceanic | Riverine | Benthic | Total |
|---|---|---|---|---|---|
| Control | DIN | 2.19 (61%) | 0.89 (25%) | 0.52 (14%) | 3.61 |
| | PHY | 0.49 (46%) | 0.38 (35%) | 0.20 (19%) | 1.07 |
| L-open ocean | DIN | 1.44 (51%) | 0.88 (31%) | 0.52 (18%) | 2.85 (-21%) |
| | PHY | 0.35 (37%) | 0.39 (41%) | 0.21 (22%) | 0.96 (-10%) |
| U-open ocean | DIN | 2.95 (67%) | 0.91 (21%) | 0.53 (12%) | 4.40 (+22%) |
| | PHY | 0.61 (52%) | 0.36 (31%) | 0.19 (16%) | 1.17 (+9.4%) |
| L-rivers | DIN | 2.16 (65%) | 0.62 (19%) | 0.52 (16%) | 3.30 (-8.6%) |
| | PHY | 0.50 (51%) | 0.28 (28%) | 0.13 (21%) | 0.99 (-7.5%) |
| U-rivers | DIN | 2.22 (52%) | 1.53 (36%) | 0.54 (12%) | 4.30 (+19%) |
| | PHY | 0.47 (39%) | 0.54 (45%) | 0.19 (16%) | 1.20 (+12%) |
| L-sediment | DIN | 2.18 (62%) | 0.89 (26%) | 0.41 (12%) | 3.49 (-3.3%) |
| | PHY | 0.50 (48%) | 0.38 (37%) | 0.16 (15%) | 1.04 (-2.8%) |
| U-sediment | DIN | 2.19 (58%) | 0.90 (24%) | 0.67 (18%) | 3.76 (+4.2%) |
| | PHY | 0.48(44%) | 0.37 (34%) | 0.25 (22%) | 1.11 (+3.7%) |


## 5 Conclusions

In this study, we investigate the behaviours of oceanic, riverine, and benthic nutrients to the inventory of nutrients and phytoplankton in the SIS using a low-trophic model with an embedded tracking module. Our study shows the largest contribution of oceanic nutrients to the inventory of nutrients and phytoplankton in the SIS and its lowest efficiency in the primary production in the SIS. On the other hand, although the riverine or benthic nutrients have a lower contribution to the inventory of nutrients and phytoplankton in the SIS than the oceanic nutrients do, they demonstrate a higher efficiency in the primary production in the SIS. It must be noted that their contributions have a strong spatial variation, in which the oceanic

nutrients range from >90% in the area close to the open ocean to <30% in the nearshore areas, while the riverine or benthic nutrients range from >50% in the nearshore areas to <10% in the areas far from the estuaries and inner bays.

The above results give us several hints for the management strategy for the SIS. Firstly, the oceanic nutrients provide a background of nutrients that supports the primary production in the SIS, and it can cause a temporal variation with a range of 20% in the inventory of nutrients and phytoplankton. This part of nutrients is mainly controlled by the variations in the nutrient concentration in the open ocean and the water exchange between the SIS and open ocean and human-made management on nutrient load from land cannot affect this part of nutrients. Secondly, the management should be easily applied to the riverine

nutrients whose impact has a strong spatial distribution. Therefore, the target of management should be carefully examined to confirm the local impacts of each river before the determination of the management plan. Thirdly, although it is a little difficult, it needs to pay more attention to the sediments, because its impacts are larger than those of riverine nutrients in some places.

### Code availability

The source code of the numerical model used in this study is available on request. Please contact the corresponding author.

**Author contribution**

JZ developed the hydrodynamic model. AM developed the low-trophic model. QL combined two models and introduced the tracking module into them. QL and XG prepared the manuscript. All the authors reviewed the manuscript.

### Competing interests

The authors declare that they have no conflict of interest.

**Acknowledgements**

This research was performed by the Environment Research and Technology Development Fund (JPMEERF20205005) of the Environmental Restoration and Conservation Agency Provided by the Ministry of Environment of Japan. We thank Dr. Yoshitsugu Koizumi who worked in the Fisheries Research Center, Ehime Prefectural Institute of Agriculture, Forestry and Fisheries, Japan and provided us with the observed nutrients and water temperature data in the Bungo Channel.

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
