# Peer review of "Contribution of open ocean to the nutrient and phytoplankton inventory in a semi-enclosed coastal sea"

_EGUsphere, 2023_

## Author Comment (AC1)

**Response to Reviewer #1**

**Always given as follows:**

**Referee comment: RC;** Author response: AR; Changes to the manuscript

**General comments**

This is a numerical model study to identify the contribution of three sources of nitrogen, which is from the land, from the open sea, and from the seafloor to DIN in a semi-enclosed coastal sea (Seto Inland Sea: SIS). Although this information is important for the prevention of eutrophication in the sea, the calculation method has some problems as shown below.

Thank you for your valuable comments on this study. We have carefully considered your comments regarding the calculation method and would like to address them in this response note.

**Specific comments**

**RC1:** Why do you include seafloor sources as a nitrogen source in addition to the open sea and terrestrial sources?

Unlike the case of phosphorus, nitrogen leached from the seafloor is the result of mineralization of "new" sediments, so nitrogen originating from the seafloor may be included in the open sea nitrogen and land nitrogen.

AR1: Thank you for your suggestion. We agree with your idea that the nitrogen leached from the seafloor is the result of the mineralization of "new" sediments. The nutrients contained in "new" sediments have an initial origin from either the rivers or the open ocean. The reason we treated it as a source is that many scientists always argue about the contribution of sediment to the nutrient in the water column. Therefore, this is just a practice way to define the nutrient source.

As we consider the sediment source of nutrients, we also realize that it has different processes from those from rivers and the open ocean. First, the processes by which particulate organic nitrogen (PON) settles onto the sediment, decomposes into dissolved inorganic nitrogen (DIN), and returns to the water column are different from the directly discharged nutrients of riverine or oceanic DIN, as well as the remineralization of PON in the water column. The sediment-released nutrients are suggested to be particularly important in shallow waters (Radtke et al., 2019). Second, there is a time lag between the deposition and mineralization of PON, either short-term or long-term (Soetaert et al., 2000). This means a large uncertainty in the proportion of nutrients released from sediment as compared to those supplied from rivers and the open ocean. Third, it is more difficult to control the nutrients released from the sediment than those supplied by rivers. Therefore, identifying the areas where the sediment-released nutrients dominate is helpful to the effective control or regulation of the riverine nutrients.

In conclusion, it is still necessary to track the sediment-released nutrients as a separate source. To address

your comment, we will mention that the nutrients leached from the seafloor have an origin from either rivers or the open ocean in Section 2.3. In addition, based on the proportion of PON flux settled on the sediment surface originating from the open ocean and rivers, we will give a quantitative estimation of the ratio of riverine and oceanic nutrients in the sediment-released nutrients in Section 4.2.

**Reference:**

Radtke, H., Lipka, M., Bunke, D., Morys, C., Woelfel, J., Cahill, B., Böttcher, M. E., Forster, S., Leipe, T., Rehder, G., and Neumann, T.: Ecological ReGional Ocean Model with vertically resolved sediments (ERGOM SED 1.0): coupling benthic and pelagic biogeochemistry of the south-western Baltic Sea, Geosci. Model Dev., 12, 275–320, https://doi.org/10.5194/gmd-12-275-2019, 2019.

Soetaert, K., Middelburg, J. J., Herman, P. M. J., and Buis, K.: On the coupling of benthic and pelagic biogeochemical models, 29, 2000.

The sentences added to the revised manuscript are:

Added in Section 2.3:

"It needs to note that the nutrients in the sediment are originally mostly from the land, or the open ocean and the sediment is a temporary storage or a permanent sink. In this study, however, we treat the sediment as the third source to track. This is because the sediment-released nutrients are gaining more attention and particularly important in shallow waters (Radtke et al., 2019)."

Added in Section 4.2:

"If the proportion of PON flux settled on the sediment surface produced by the oceanic and riverine nutrients were taken as the proportion of DIN flux released from the sediment, among the 86 mol N s$^{-1}$ of DIN from the sediment, 60 mol N s$^{-1}$ has an origin from the open ocean and 26 mol N s$^{-1}$ from rivers."

Radtke, H., Lipka, M., Bunke, D., Morys, C., Woelfel, J., Cahill, B., Böttcher, M. E., Forster, S., Leipe, T.,

**RC2:** Why is dissolved organic nitrogen not included in the calculation?

Dissolved organic nitrogen, which accounts for about 90% of the total nitrogen in SIS, is not included in the calculation. The Ministry of the Environment's total load reduction for SIS is also based on total nitrogen in its calculations.

**AR2:** Thank you for your comment. Each biogeochemical model is designed for a different purpose, and the number and type of variables and biogeochemical processes are selected depending on the purpose (Fennel et al., 2022). The introduction of more variables and processes will introduce more uncertainties and calculation costs. In addition to expressing biogeochemical processes explicitly, expressing them implicitly is a substitute. The purpose of this study is to evaluate the inventories of nutrients and phytoplankton originating from different sources in the SIS. The direct transformations between them are of the most concern. Therefore, the widely used NPZD model including DIN, PHY, ZOO, and PON, is sufficient for the research purpose. For example, to estimate the nitrogen fluxes in the shelf area of the Middle Atlantic Bight, Fennel et al. (2006) also used an NPZD model to represent the nitrogen processes in the water column.

Although the research about the dissolved organic nitrogen (DON) in SIS was not often reported and the dataset of DON was also little, we still collected some results. In Hiuchi Nada and Iyo Nada, which are two sub-regions of SIS, Kumamoto et al. (1994) estimated the DON concentration at two sampling stations in late spring. They reported that DON concentration was superior to DIN concentration and nearly constant in the vertical and showed little temporal variation in Iyo Nada. By subtracting DIN from TN, Asahi et al. (2019) obtained the DON concentration at the surface layer of SIS in the 1990s and 2000s. They reported that DON concentration was higher than DIN and was relatively constant. Because we are interested in the materials that were dynamically changed, we did not include the little changed DON in our model. In other words, DON is like a background for TN but only DIN has a strong relationship with the low-trophic ecosystem. After reading your comment, however, we acknowledge that the incorporation of DON can be considered for some specific questions if there is a sufficient amount of observational DON to support model construction.

**Reference:**

Asahi T., Abo K., Abe K., and Tada K.: Comparison of Dissolved Inorganic and Organic Nitrogen between the sand s in the Seto Inland Sea of Japan, Bulletin on Coastal Oceanography, 56, 123–131, 2019.

Kumamoto, Y., Tsubota, H., and Fujiwara, K.: Temporal Variation of Dissolved Organic Nitrogen and Phosphorus, Environmental Science, 7, 1–12, https://doi.org/10.11353/sesj1988.7.1, 1994.

**RC3:** Are the DIN boundary conditions used in the numerical model reasonable?

The boundary condition is that the open sea origin DIN is zero at the seafloor and landward.

If there are no biochemical processes in SIS, no DIN supply from land or seafloor, only physical diffusion, then the DIN concentration in SIS is equal to the open boundary DIN (DIN from the open sea) and SIS is filled with DIN from the open sea. In other words, DIN = 0 does not occur on the seafloor surface or the landward shore.

**AR3:** Yes, the DIN boundary conditions used in the numerical model and the tracking cases are reasonable. From the mathematical perspective, the numerical model is represented as a system of coupled partial differential equations. The tracking method we used in this study is a linear decomposition of the original partial differential equations. In order to solve the subset of partial differential equations, the boundary conditions also need to be linearly decomposed. For the boundary condition of the tracking open ocean case, we specified the open ocean origin DIN zero at the land boundary and seafloor. This is because the open ocean origin DIN is not released into the SIS from the land or seafloor. The case you said that SIS is filled with DIN from the open sea is the analytical solution of a diffusion equation with zero flux from land and a fixed concentration at the open ocean side. We actually used the same boundary conditions as the case you said. In our results, the open ocean-origin DIN is not zero at the landward shore or on the seafloor surface (Fig. 5b and Fig. 5e).

**Technical correction**

**RC4**: It should be noted that the land load in this report is an underestimate.

The terrestrial nitrogen load for SIS is published every five years by the Ministry of the Environment of Japan (MEJ). It is necessary to state the values of the terrestrial load by MEJ and the terrestrial load in this report.

In SIS, which experienced eutrophication in the 1970s, the majority of domestic and industrial wastewater is treated at treatment facilities on the waterfront and discharged directly into the sea in recent years. Therefore, there is a large difference between the DIN flow via rivers and the total nitrogen flow actually entering the sea (especially in the eastern Seto Inland Sea).

In SIS, river discharge is significantly lower in winter, resulting in large seasonal variations in DIN flow from rivers, whereas there is little seasonal variation in DIN flow from domestic and industrial sources. This affects the seasonal variation of DIN concentration in the SIS.

**AR4:** Thank you for your suggestions. We have recognized that the DIN load from the land was underestimated in our study, and we mentioned it on Line 115 to Line 117. In the revised manuscript, we will appropriately increase the DIN load from the land using the total nitrogen (TN) load for SIS published every five years by the Ministry of the Environment of Japan, and we will give a quantitative estimate of how large this underestimation in the DIN load from land is. In this response letter, we first report the related information we have collected.

TN loads from land to the SIS were estimated by the Ministry of the Environment, Japan every five years from 1979 based on the unit load method in the catchment area of SIS (Abo and Yamamoto, 2019; Timita et al., 2016). From 1979 to 2014, the average TN load from land to the SIS was 471 mol N s$^{-1}$. Yanagi and Ishii (2004) indicated that the TN loads estimated by the unit load method did not reproduce the inflow of TN to the coastal sea since some parts of TN loads remained on land. Yamamoto et al. (1996) recommended the use of river flow rate to calculate the actual inflow TN load into the SIS and reported that the TN load calculated using this method was about 48% of that measured by the unit load method. Based on this value presented by Yamamoto et al. (1996), the average inflow TN load to the SIS was 226 mol N s$^{-1}$, which flowed into the SIS from the land through the 21 first-order rivers and about 640 other small rivers. We included 21 first-order rivers and 45 small rivers in our study. In addition, the compounds of TN loads from land are not clear. The proportion of DIN concentration in TN concentration at the first-order rivers of SIS is about 77%, which was estimated by the nutrient data from the Ministry of Land, Infrastructure, Transport and Tourism, Japan (http://www1.river.go.jp/). If we apply this value to the TN load from land, the DIN load from land is 174 mol N s$^{-1}$, which is 2.7 times higher than the DIN load from rivers (64 mol N s$^{-1}$) estimated by our study. A more extreme situation is that in addition to DIN, other compounds of TN can also be used by the phytoplankton through complex biogeochemical processes. Then our estimation of 64 mol N s$^{-1}$ will be 28% of the TN load from land. In order to consider the DIN load from land as much as possible in revision, a new series of experiments is conducted to increase the DIN load from the rivers to 3 times the original value (64 mol N s$^{-1}$) to represent the DIN load from land. These experiments are being calculated and the new results

will be described in the revised manuscript.

**Reference:**

Abo, K. and Yamamoto, T.: Oligotrophication and its measures in the Seto Inland Sea, Japan, Bulletin of Japan Fisheries Research and Education Agency, 49, 21–26, 2019.

Tomita, A., Nakura, Y., and Ishikawa, T.: New direction for environmental water management, Marine Pollution Bulletin, 102, 323–328, https://doi.org/10.1016/j.marpolbul.2015.07.068, 2016.

Yamamoto, T., Kitamura, T., and Matsuda, O.: Riverine inputs of fresh water, total nitrogen and total phosphorus into the Seto Inland Sea, Journal of the Faculty of Applied Biological Science, Hiroshima University, 35, 81–104, 1996.

Yanagi, T. and Ishii, D.: Open Ocean Originated Phosphorus and Nitrogen in the Seto Inland Sea, Japan, J Oceanogr, 60, 1001–1005, https://doi.org/10.1007/s10872-005-0008-4, 2004.

**RC5:** Section 3.2. It is important to indicate the time required for the numerical model to become stationary; the DIN flow path during the set-up period is not the flow path when the model becomes stationary.

**AR5:** Thank you for your suggestions. We indicated the time required for the numerical model to become stationary in Section 2.2. In Lines 150 to 151: "The hydrodynamic-biogeochemical model was initiated on the first day of January and stabilized from the third year onwards. Therefore, the simulation results of the third year were used to analyze the seasonal variations of DIN and PHY.". For the tracking case, at Line 171: "We initiated the tracking technique from the first day of the fourth year of the hydrodynamic-biogeochemical model, …" and at Line178 to 179: "After a spin-up of three years, the annual cycle of each source of nutrients and related particles became stationary.". In order to make readers clear about the time required for the numerical model to become stationary, we will add one figure to describe the ratio of (DIN_ocean+DIN_river+DIN_sediment)/DIN in the SIS from the first year to the third year of the tracking simulation in the revised Supplement Materials.

The purpose of Section 3.2 is to exhibit the pathway of oceanic, riverine, and benthic DIN gradually occupying the SIS from the initial state. After one year of calculation, DIN concentrations from the open ocean, rivers, and sediment have already occupied most areas of the SIS. Therefore, we presented the results of the first year of tracking cases in Fig. 4 to depict this pathway.

**RC6:** Actual measurements of the amount of nitrogen and phosphorus entering SIS from the open sea were made by several organizations in the 1980s to 2000s, and it has been shown that the amount of nitrogen entering from the open sea is equivalent to the amount of land-based load during the summer months. It is desirable to cite these papers.

**AR6:** Thank you for your suggestions. In revision, we collected the related papers and present them in this response note. We will cite these papers in Section 4.2 of the revised manuscript.

Fujiwara et al. (1997) reported that about 28 mol s$^{-1}$ of TN was transported from the open ocean to the SIS

through Bungo Channel based on observations during 15 days from July to August in 1982 and 140 mol s$^{-1}$ of TN through Kii Channel from open ocean based on observations during 2 days in August of 1985. Kasai et al. (2001) estimated the net DIN transport through Kii Channel into SIS in August of 1996 was 111 mol s$^{-1}$ which was similar to that of Fujiwara et al. (1997). However, they also report that in Augusts of 1997 and 1999, only 6.5 mol s$^{-1}$ of DIN and 37 mol s$^{-1}$ of DIN were transported into the SIS, respectively.

By subtracting the amount of DIN load buried in the sediment from the amount of DIN load from land, Fujiwara et al. (2006) reported that 50 mol s$^{-1}$ of TN was transported from the open ocean to the SIS.

Although quantitative estimates of DIN transport from the open ocean were not available, Takashi et al. (2002) revealed a strong (weak) inflow of DIN transport to the SIS when the Kuroshio is offshore (inshore) in summer, but the outflow in winter was regardless of the path of Kuroshio based on monthly observations from April 1999 to December 2001.

**Reference:**

Fujiwara, T., Uno, N., Tada, M., Nakatsuji, K., Kasai, A., and Sakamoto, W.: Inflow of nitrogen and phosphorus from the ocean into the Seto Inland Sea, Proc. Coastal Engineering (JSCE), https://doi.org/10.2208/proce1989.44.1061, 1997.

Fujiwara, T., Kobayashi, S., Kunii, M., and Uno, N.: Nitrogen and phosphorus in Seto Inland Sea: Their origin, budget and variability, Bull Coast Oceanogr, 43, 129–136, 2006.

Kasai, A., Fujiwara, T., and Tada, M.: Ocean Structure and Nutrient Transport in the Kii Channel, Proceedings of Coastal Engineering, 48, 436–440, https://doi.org/10.2208/proce1989.48.436, 2001.

Takashi, T., Fujiwara, T., Sumitomo, T., and Takeuchi, J.: Transport of nitrogen and phosphorus from the open ocean to the Kii Channel, Proceedings of Coastal Engineering, 49, 1076–1080, https://doi.org/10.2208/proce1989.49.1076, 2002.

**RC7:** Line 114: It should be noted that the seasonal variation of the nitrogen load from rivers is due to the seasonal variation of the river flow. Unlike Europe, the SIS receives a little precipitation in winter.

**AR7:** Thank you for your suggestions. We will note this information in the revised manuscript.

The revised expression is:

"The seasonal variation in DIN loads of all rivers, with high loads in July and September and low loads in January, is primarily controlled by the seasonal variation in river discharge. The annual mean of DIN loads from rivers is 63.85 mol s$^{-1}$."

---

## Author Comment (AC2)

**Response to Reviewer #2**

**Always given as follows:**

**Referee comment: RC;** Author response: AR; Changes to the manuscript

**General comments**

**RC1:** Information about the hydrodynamics of the SIS is missing, specifically the main currents should be described and compared between your model and e.g., literature references, since it is essential that the advective transport is realistically captured, which is not obvious from validating the DIN and DIP concentrations alone. Also comparing salinity to observations might help to check whether the mixing ratio between riverine and oceanic water masses is realistically captured in the model.

**AR1:** Thank you for your suggestions. The hydrodynamic model used in this study was the same as that in Chang et al. (2009) and Zhu et al. (2019). They have finished the general comparisons with observations (residual current pattern, monthly water temperature, and monthly salinity) for the hydrodynamic model, which confirmed that this model can generally reproduce the major hydrodynamic characteristics of the Seto Inland Sea (SIS). In addition, this model has been used to study the formation of cold bottom water and some related processes (Yu et al., 2016; Yu and Guo, 2018) as well as to calculate the water age of river water (Wang et al., 2018). Therefore, we only cited Chang et al. (2009) and Zhu et al. (2019) in the original manuscript but did not give a detailed description, which is shown as follows:

"Chang et al. (2009) compared the simulated surface residual current of this SIS hydrodynamic model with the observations. It showed that the summer and winter circulation patterns were reproduced. Significant cyclonic and anticyclonic eddies were developed near the entrance and inner part of Suo Nada, respectively, both in the simulated and observed results. In addition, the model also captured the southward current flowing to the western Bungo Channel and the southwestward current in the northern Iyo Nada, which were also evident in the observations. The model also well reproduced the observed circulation features in the Harima Nada."

"Zhu et al. (2019) compared the simulated temperature and salinity of SIS in February (winter) and July (summer) with the observations. They reported that the warm and saline waters flowed into the SIS through the Bungo Channel and Kii Channel in winter, which was consistent with the observations. For the vertical distributions in winter, both the simulated and observed results showed that the water column was well mixed throughout the whole SIS. In summer, both temperature and salinity exhibited a well-mixed pattern around the straits and a well-stratified pattern in the broad basins. Low salinity existed in Osaka Bay, forming a front structure with high salinity water in the Kii Channel."

In the revised manuscript, based on your suggestions, we will describe the main current field of SIS and compare the model results with the literature in Section 3.1. For the salinity distribution, we have obtained a long-term monthly observation dataset carried out by the prefectural fishery research centers around the SIS. We will also compare the simulated salinity results with observations in Section 3.1 of the revised

manuscript.

**Reference:**

Chang, P.-H., Guo, X., and Takeoka, H.: A numerical study of the seasonal circulation in the Seto Inland Sea, Japan, J. Oceanogr., 65, 721–736, https://doi.org/10.1007/s10872-009-0062-4, 2009.

Wang, H., Guo, X., and Liu, Z.: The age of Yodo River water in the Seto Inland Sea, Journal of Marine Systems, 191, 24–37, https://doi.org/10.1016/j.jmarsys.2018.12.001, 2019.

Yu, X. and Guo, X.: Intensification of water temperature increase inside the bottom cold water by horizontal heat transport, Continental Shelf Research, 165, 26–36, https://doi.org/10.1016/j.csr.2018.06.006, 2018.

Yu, X., Guo, X., and Takeoka, H.: Fortnightly Variation in the Bottom Thermal Front and Associated Circulation in a Semienclosed Sea, Journal of Physical Oceanography, 46, 159–177, https://doi.org/10.1175/JPO-D-15-0071.1, 2016.

Zhu, J., Guo, X., Shi, J., and Gao, H.: Dilution characteristics of riverine input contaminants in the Seto Inland Sea, Mar. Pollut. Bull., 141, 91–103, https://doi.org/10.1016/j.marpolbul.2019.02.029, 2019.

**RC2**: You state that your nitrogen loads are smaller than previously reported values, as you neglect industrial and land-based sources as well as particulate forms of riverine nitrogen. It seems you also ignore atmospheric deposition? Since the main result of the paper, which is the oceanic fraction of the nutrients in the ecosystem, will be strongly dependent on the terrestrial loads you put in, please give a quantitative estimate on how large this uncertainty/error in the loads is.

**AR2:** Thank you for your comment.

**First question:** Yanagi (1997) considered the deposited total nitrogen (TN) load in rainwater when estimating the nitrogen budget in SIS. In this estimation, the net TN load of atmospheric deposition was 8% of the land input. In a coastal area of SIS during the spring of 2015, the dry deposition fluxes of particulate $NH_4$ and $NO_3$ were $2.3 \times 10^{-7}$ mol m$^{-2}$ s$^{-1}$ and $5.5 \times 10^{-7}$ mol m$^{-2}$ s$^{-1}$, respectively (Nakamura et al., 2020). These atmospheric aerosols were measured on a rooftop at Kagawa College, Kagawa Prefecture, Japan. We are not sure whether they can represent the atmospheric aerosols for the whole SIS. Although there is great uncertainty, we still apply these two values to the whole SIS with an area of 23,203 km$^2$. The estimated dry deposition fluxes of particulate $NH_4$ and $NO_3$ for the SIS were 5.4 mol N s$^{-1}$ and 12.7 mol N s$^{-1}$, respectively. They are lower than the nitrogen input from rivers, the open ocean, and sediment (64 mol N s$^{-1}$, 174 mol N s$^{-1}$, 86 mol N s$^{-1}$). This is the result of spring and there is no study for other seasons. Considering these uncertainties, we did not include the atmospheric deposition for the SIS.

**Second question:** TN load from land to the SIS was estimated by the Ministry of the Environment, Japan every five years from 1979 based on the unit load method in the catchment area of SIS (Abo and Yamamoto, 2019; Timita et al., 2016). From 1979 to 2014, the average TN load from land to the SIS was 471 mol N s$^{-1}$. Yanagi and Ishii (2004) indicated that the TN load estimated by the unit load method did not reproduce the inflow of TN to the coastal sea since some parts of the TN load remained on land. Yamamoto et al. (1996)

recommended the use of river flow rate to calculate the actual inflow TN load to the SIS and reported that the TN load calculated using this method was about 48% of that measured by the unit load method. Based on this value presented by Yamamoto et al. (1996), the average inflow TN load to the SIS was 226 mol N s$^{-1}$, which flowed into the SIS from the land through the 21 first-order rivers and about 640 other small rivers. We included 21 first rivers and 45 small rivers in our study. In addition, the compounds of TN load from land are not clear. The proportion of dissolved inorganic nitrogen (DIN) concentration in TN concentration at the first-order rivers of SIS is about 77%, which was estimated by the nutrient data from the Ministry of Land, Infrastructure, Transport and Tourism, Japan (http://www1.river.go.jp/). If we apply this value to the TN load from land, the DIN load from land is 174 mol N s$^{-1}$, which is 2.7 times higher than the DIN load from rivers (64 mol N s$^{-1}$) estimated by our study. A more extreme situation is that in addition to DIN, other compounds of TN can also be used by the phytoplankton through complex biogeochemical processes. Then, our estimation of 64 mol N s$^{-1}$ will be 28% of the TN load from land. To consider the DIN load from land as much as possible, a new series of experiments is conducted to increase the DIN load from the rivers to 3 times its original value (64 mol N s$^{-1}$) to represent the DIN load from land. These experiments are being calculated and the new results will be described in the revised manuscript.

**Reference:**

Abo, K. and Yamamoto, T.: Oligotrophication and its measures in the Seto Inland Sea, Japan, Bulletin of Japan Fisheries Research and Education Agency, 49, 21–26, 2019.

Nakamura, T., Narita, Y., Kanazawa, K., and Uematsu, M.: Organic Nitrogen of Atmospheric Aerosols in the Coastal Area of Seto Inland Sea, Aerosol Air Qual. Res., 20, 1016–1025, https://doi.org/10.4209/aaqr.2019.12.0658, 2020.

Tomita, A., Nakura, Y., and Ishikawa, T.: New direction for environmental water management, Marine Pollution Bulletin, 102, 323–328, https://doi.org/10.1016/j.marpolbul.2015.07.068, 2016.

Yamamoto, T., Kitamura, T., and Matsuda, O.: Riverine inputs of fresh water, total nitrogen and total phosphorus into the Seto Inland Sea, Journal of the Faculty of Applied Biological Science, Hiroshima University, 35, 81–104, 1996.

Yanagi, T.: Budgets of fresh water, nitrogen and phosphorus in the Seto Inland Sea, Umi-no-Kenkyu, 6, 157–161, 1997.

Yanagi, T. and Ishii, D.: Open Ocean Originated Phosphorus and Nitrogen in the Seto Inland Sea, Japan, J Oceanogr, 60, 1001–1005, https://doi.org/10.1007/s10872-005-0008-4, 2004.

**RC3:** Sediment DIN flux: Your sediment model is very simplistic and maybe a bit too simplistic for your application. You assume constant DIN fluxes from the sediments in a study where you state that your goal is to understand the temporal dynamics of eutrophication. You ignore a positive feedback loop in which enhanced nutrient loads lead to more settling PON, to higher reactive TN concentrations in the surface sediment and subsequently to higher DIN release from the sediments. Please at least discuss the potential

implications of this strong simplification in your discussion section. This is especially critical since sediment-water DIN fluxes are not easily observable. They tend to show substantial small-scale variation depending on e.g. the presence of bioturbating or bioirrigating macrofauna. Please give more information on what the uncertainty of the benthic flux estimates is.

**AR3:** Thank you for your suggestions. In our study, the DIN flux from the sediment was calculated by the surface sediment TN concentration and bottom temperature based on an empirical function (Tada et al., 2018). This empirical function is based on the measured DIN flux in the laboratory using the sediment collected in many stations in the SIS (Tada et al., 2018). The reason we used it is because it reflects the real situation in the SIS.

As we applied this formula in this study, we used the mean sediment surface TN concentration averaged from the observation data in the past 40 years, which reflected only an average state of surface TN concentration over this period and therefore ignored its long-term trend. Because the surface sediment TN concentration used in the formula is independent of the particle flux from the water column, it has not the feedback dynamic you mentioned. In fact, the only temporal variation in the DIN flux from the sediment in this study was induced by the annual variations of the bottom temperature derived from the hydrodynamic model.

Because we included the process of resuspension of particles from the sediment surface and its decomposition in the water column, we think that the short-term effects of PON settled to the sediment surface have been treated as a part of the nitrogen cycle processes in the water column. On the other hand, we also feel that it is not reasonable to treat the DIN flux from such short-term effects as a source of nutrients. In other words, our benthic nutrient flux reflects only the long-term one whose timescale is close to one year.

To make these points a little clear, we will add some sentences in the revised manuscript.

**Reference:**

Tada K., Nakajima M., Yamaguchi H., Asahi T., and Ichimi K.: The Nutrient Dynamics and Bottom Sediment in Coastal Water, Bull. Coast. Oceanogr., 55, 113–124, https://doi.org/10.32142/engankaiyo.55.2_113, 2018.

The revised expression is:

"It should be noted that the calculation of DIN flux released from the sediment is somewhat simple in our model. We used the annual mean TN concentration and bottom temperature based on an empirical function to calculate the sediment DIN flux. This means that we did not consider the instant effect of particulate organic nitrogen (PON) settled to the surface sediment, which can increase the reactive TN concentration in the surface sediment and subsequently higher DIN flux released from the sediment. In fact, it is difficult to treat such short-term responses of benthic DIN flux to the settled PON (Soetaert et al., 2000) as a source of nutrients because they can be a part of the nutrient cycle within the water column."

Soetaert, K., Middelburg, J. J., Herman, P. M. J., and Buis, K.: On the coupling of benthic and pelagic biogeochemical models, 29, 2000.

**RC4:** You consider sedimentary DIN as a "source". Actually, sediments are not a source for nutrients, but just a temporary storage or a permanent sink. The nutrients stored in the sediment are originally mostly from riverine or oceanic origin. Even if this may be somehow clear for most readers, I think it is still worth mentioning.

**AR4:** Thank you for your suggestions. We agree with your view. We will mention this information in Section 2.3. In addition, based on the proportion of PON flux settled on the sediment surface originating from the open ocean and rivers, we will give a quantitative estimation of the ratio of riverine and oceanic nutrients in the sediment-released nutrients in Section 4.2.

The sentences added to the revised manuscript are:

Section 2.3:

"It needs to note that the nutrients in the sediment are originally mostly from the land, or the open ocean and the sediment is a temporary storage or a permanent sink. In this study, however, we treat the sediment as the third source to track. This is because the sediment-released nutrients are gaining more attention and are particularly important in shallow waters (Radtke et al., 2019)."

Section 4.2:

"If the proportion of PON flux settled on the sediment surface produced by the oceanic and riverine nutrients were taken as the proportion of DIN flux released from the sediment, among the 86 mol N $s^{-1}$ of DIN from the sediment, 60 mol N $s^{-1}$ has an origin from the open ocean and 26 mol N $s^{-1}$ from rivers."

Radtke, H., Lipka, M., Bunke, D., Morys, C., Woelfel, J., Cahill, B., Böttcher, M. E., Forster, S., Leipe, T.,

**RC5:** Another point maybe worth discussing is that the "oceanic" DIN can be of riverine origin, just added to the Japanese coastal waters from rivers outside the SIS. Or is it the "open" Pacific Ocean signal that is really controlling the conditions at the borders of your model domain?

**AR5:** Thank you for your comment. First, the DIN concentration specified at the open boundaries of our model domain was derived from the relationship between the observed water temperature and DIN concentration south of the open boundaries, which was provided in the Supplement Materials. Their strong correlation at the range of lower temperatures reflects the inherent nature of water temperature and DIN concentration of the Kuroshio subsurface water. At the range of higher temperatures, the DIN concentration is low, which reflects the nutrient-poor Kuroshio surface water.

Second, some first-order rivers are flowing into the coastal waters of Kyushu, west of Japan. In principle, these river waters can pass the Bungo Channel and Kii Channel. However, as these waters reach the areas outside the SIS, they have been largely diluted by the Kuroshio. As we know, the river discharge is at an order of several hundreds of $m^3$ $s^{-1}$ while the Kuroshio has a volume transport of several tens of $10^6$ $m^3$ $s^{-1}$. Furthermore, it needs more than one month for these waters to reach the areas outside the SIS and therefore most of the nutrients from the rivers have been used by the phytoplankton in the pathway.

For the above reasons, we concluded that the open boundary conditions really reflect the signals of the open

ocean (Pacific Ocean). To make it clear to the readers, we will give more information in Section 2.2 of the revised manuscript.

**RC6:** Please give some references why it is reasonable to exclude dinitrogen fixation as a relevant N source in the SIS and neglect it in the model. (in other coastal seas it is a majour source)

**AR6:** Thank you for your comment. There are few studies about dinitrogen fixation for the whole SIS. In Osaka Bay, which is a severely polluted sub-region of eastern SIS, Hashimoto et al. (2016) reported a nitrogen fixation of 0.0011 mol N s$^{-1}$ using the nitrogen fixation rate and cell abundance of unicellular diazotrophic cyanobacteria. This value was much lower than the nitrogen input of rivers into Osaka Bay (~19 mol N s$^{-1}$, Fig. S2). Lee et al. (1996) reported that there was no nitrogen fixation observed in Hiroshima Bay. Based on Lee et al. (1996), Yamamoto et al. (2008) assumed no nitrogen fixation in the whole SIS when estimating the nitrogen budget for the SIS. According to these studies, we think it is reasonable to exclude nitrogen fixation as a relevant N source in the SIS and neglect it in the model. In the future, we will include nitrogen fixation in the model if there are more observations available.

**Reference:**

Hashimoto, R., Watai, H., Miyahara, K., Sako, Y., and Yoshida, T.: Spatial and temporal variability of unicellular diazotrophic cyanobacteria in the eastern Seto Inland Sea, Fish Sci, 82, 459–471, https://doi.org/10.1007/s12562-016-0983-y, 2016.

Lee, Y. S., Seiki, T., Mukai, T., Takimoto, K., and Okada, M.: Limiting nutrients of phytoplankton community in Hiroshima Bay, Japan, Water Research, 30, 1490–1494, 1996.

Yamamoto, T., Hiraga, N., Takeshita, K., and Hashimoto, T.: An estimation of net ecosystem metabolism and net denitrification of the Seto Inland Sea, Japan, Ecological Modelling, 215, 55–68, https://doi.org/10.1016/j.ecolmodel.2008.02.034, 2008.

**RC7:** Section 4.2 is lacking information on how the figures presented in the article relate to previous estimates of the nitrogen budget of the SIS.

**AR7:** Thank you for your suggestions. We have collected some related information and will present it in this response note. We will also add this information in Section 4.2 of the revised manuscript.

Yanagi (1997) estimated the nitrogen budget in the SIS based on some assumptions. It reported that 392 mol N s$^{-1}$ of TN was transported from the land and 31 mol N s$^{-1}$ was deposited by rainwater. 358 mol N s$^{-1}$ of TN was transported to the open ocean at the open boundaries, and 64 mol N s$^{-1}$ was buried in the sediment. Our study revealed that 64 mol N s$^{-1}$ of DIN was from rivers, among which 14 mol N s$^{-1}$ was transported at the open boundaries and 50 mol N s$^{-1}$ was buried in the sediment. Since we did not introduce the other types of nitrogen from rivers, our values are much lower than those reported by Yanagi (1997).

Fujiwara et al. (2006) also estimated the TN budget in the SIS and clarified the land origin and open ocean origin, showing that 330 mol N s$^{-1}$ of TN was supplied from the land to SIS, of which 297 mol N s$^{-1}$ was

transported to the open ocean and 33 mol N s$^{-1}$ was buried to the sediment. Fujiwara et al. (2006) also reported that the net input of TN from the open ocean was 50 mol N s$^{-1}$, which was buried in the sediment. In this study, the TN originating from the open ocean has a net input of 62 mol N s$^{-1}$, all of which was buried in the sediment.

Compared our study with the above two studies, the main difference was the amount of TN from the land. Even though they made some adjustments for the TN obtained from the original unit method calculations, the estimates given based on experience have a high degree of uncertainty and were not linked to river discharges. Fujiwara et al. (2006) also stated that TN from the land they estimated had a great deal of uncertainty. According to our answer to RC2, we believe that it is more accurate to combine the river flow and the DIN load occurring on land to give the actual load flowing into SIS in the revised manuscript.

There are also studies to estimate the DIN transport at boundaries between the SIS and the open ocean. At the south of Bungo Channel (the west open boundary of our model), Morimoto et al. (2022) reported that a net of 385 mol N s$^{-1}$ oceanic DIN was transported from the open ocean to the SIS in July and August based on simulated water volume and DIN concentration derived from water temperature. It was 245 mol N s$^{-1}$ in our study. The reason that our estimate is less than theirs may be caused by the outward DIN transport. We used the DIN concentration calculated by the low-trophic ecosystem model, which was larger than their DIN concentration. Fujiwara et al. (1997) reported about 168 mol N s$^{-1}$ of DIN through Kii Channel from the open ocean in August of 1985. We estimated about 139 mol N s$^{-1}$ of DIN was from the open ocean to the SIS through Kii Channel in August.

In revision, we will add the above information to the manuscript.

**Reference:**

Fujiwara, T., Uno, N., Tada, M., Nakatsuji, K., Kasai, A., and Sakamoto, W.: Inflow of nitrogen and phosphorus from the ocean into the Seto Inland Sea, Proc. Coastal Engineering (JSCE), https://doi.org/10.2208/proce1989.44.1061, 1997.

Fujiwara, T., Kobayashi, S., Kunii, M., and Uno, N.: Nitrogen and phosphorus in Seto Inland Sea: Their origin, budget and variability, Bull Coast Oceanogr, 43, 129–136, 2006.

Morimoto, A., Dong, M., Kameda, M., Shibakawa, T., Hirai, M., Takejiri, K., Guo, X., and Takeoka, H.: Enhanced Cross-Shelf Exchange Between the Pacific Ocean and the Bungo Channel, Japan Related to a Heavy Rain Event, Front. Mar. Sci., 9, 869285, https://doi.org/10.3389/fmars.2022.869285, 2022.

Yanagi, T.: Budgets of fresh water, nitrogen and phosphorus in the Seto Inland Sea, Umi-no-Kenkyu, 6, 157–161, 1997.

**RC8:** Section 4.3 occurs very unexpectedly. If nutrient load reduction experiments are performed, this should be mentioned in the methods section and the results section and not appear for the first time in the discussion section. Anyway, the model with its assumed constant sedimentary N fluxes seems not appropriate for nutrient load scenarios, since here the sediment feedback is essential. Your model implicitly assumes that as

soon as some riverine N reaches the sediment in particulate form, its influence is gone. In reality, specifically in shallow near-coastal sediments, fresh organic matter that reaches the sediment can me remineralized quickly and (in case that this does not happen due to denitrification) become available for primary production again. So maybe leave just leave out this section (it adds a side-story to the main story line of the article) or move it to the online supplement?

**AR8:** Thank you for your suggestions. Yes, Section 4.3 is given a little unexpectedly. We will mention these sensitive experiments in the Methods section of the revised manuscript. The purpose of these sensitivity experiments is to examine the uncertainty of model results due to the change in the input flux of each source of nutrients. The processes you mentioned about the fresh matter that reaches the sediment and is remineralized quickly can be understood to be included in the nitrogen cycle within the water column in our model. This is because we introduce the resuspension processes in our model. If the bottom stress is over a critical value, the particles that reach the bottom will be returned to the water immediately. Then they will be remineralized quickly in the water column.

Again, this is also related to the definition of sediment source of nutrients. In our study, we do not want to treat such quickly remineralized nutrients as the sediment source. In our early calculation, we treated the quickly remineralized nutrients as the sediment source but found that the sediment source of nutrients became over 80% in most areas. Therefore, such bottom-touched particles were not allowed to be the sediment source of nutrients.

In the revision, we will add some sentences to explain the above points.

**Minor comments**

**RC9:** Line 30: "regulated" -> "influenced"? (Climate change has no "regulating" effect)

**AR9:** Agree. We will correct this in the revised manuscript.

**RC10:** Line 33: "presenting a different seasonal variation" -> "so their import has a seasonality that is different."

**AR10:** Agree. We will correct this in the revised manuscript.

**RC11:** Line 56: Abbreviation "COD" is not defined.

**AR11:** We will add its definition as "Chemical Oxygen Demand" in the revised manuscript.

**RC12:** Line 58: "concern about oligotrophication was raised for it" is unclear, please rephrase.

**AR12:** We will rephrase this in the revised manuscript.

The revised expression will be: "…raised concerns of oligotrophication.".

**RC13:** Line 59: meaning of "As the first step" is unclear. Are you doing a multi-step approach, or do you indicate that you are the first who try to understand these changes?

**AR13:** We mean there are several steps to understanding the long-term change in the nutrient concentrations in the SIS. In this study, we conducted the climatological simulation to quantitatively evaluate the inventory of materials originating from the open ocean, river, and sediment. In the future study, we will conduct simulations for yearly and interannual variations to figure out the long-term variation of impacts of the open ocean, rivers, and sediment.

To avoid misunderstanding, we will modify this sentence in the revised manuscript.

The revised expression is:

"To initiate our understanding such long-term change in the nutrient concentrations in the SIS, …".

**RC14:** Line 91: "from a daily dataset" is too unspecific, please give a few more details.

**AR14:** We will give more details about the daily dataset in the revised manuscript.

The revised expression is:

"…from the daily Grid Point Value of Meso-Scale Model (GPV-MSM) (http://www.jmbsc.or.jp/jp/online/file/f-online10200.html) provided by the Japan Meteorological Agency."

**RC15:** Line 93: Please specify where your hydrodynamic boundary conditions come from.

**AR15:** We will specify them in the revised manuscript.

The revised expression is:

"The open boundary conditions including de-tided current velocity, temperature, and salinity were based on the model results of Guo et al. (2004).".

**RC16:** Line 112: "The spatial variation" -> "Spatial variation"

**AR16:** We will correct it in the revised manuscript.

**RC17:** Line 133: Wang 2002 actually only cites the method from Ariathurai and Krone (1976), please give the original reference.

**AR17:** We will correct it and add this literature in the Reference.

The revised expression is:

"…we followed the method proposed by Ariathurai and Krone (1976) …".

"Ariathurai, R. and Krone, R. B.: Mathematical modelling of sediment transport in estuaries, in: Estuarine

Processes, Elsevier, 98–106, 1976.".

**RC18:** Line 172-177: Please state more clearly which fluxes you define at the boundaries. You state you define "zero concentration" but that is puzzling. At the land-sea and sediment-water boundaries you should have identical fluxes as for DIN for one of the tagged state variables and zero flux for the others. For the open boundary condition, this should be the same during times of inflow, but during times of outflow (in the upwind scheme) the DIN_??? should be exported according to the ratio DIN_???/DIN. Please clarify.

**AR18:** We will state more clearly the fluxes at the boundaries in the revised manuscript. Because we solve the DIN of each source, we have their value at the grid next to the open boundary. Therefore, we do not need to use the ratio of DIN_???/DIN to determine the flux for outflow, although its effect is the same as using the ratio.

The revised expression is:

"For the open boundary conditions, during the time of inflow, $DIN_{ocean}$ flux had the same values as those used at the open boundaries of the hydrodynamic-biogeochemical model; during the time of outflow, $DIN_{ocean}$ flux was given by the product of $DIN_{ocean}$ and the outflow velocity. $DIN_{ocean}$ flux was specified to zero at the land-sea interface and the water-sediment interface. $DIN_{river}$ at the land-sea interface was identical to those used in the hydrodynamic-biogeochemical model, but it was set to zero at the water-sediment interface. At the open boundaries, during the time of inflow, $DIN_{river}$ flux was set to zero and during the time of outflow, $DIN_{river}$ flux was given by the product of $DIN_{river}$ and the outflow velocity. $DIN_{sediment}$ was set to have the same flux at the sediment-water interface as that in the hydrodynamic-biogeochemical model, but it was set to have zero flux at the land-sea interface. At the open boundaries, during the time of inflow, $DIN_{sediment}$ flux was set to zero and during the time of outflow, $DIN_{sediment}$ flux was given by the product of $DIN_{sediment}$ and the outflow velocity."

**RC19:** Line 186: Why do you use observations in 50 m depth as "bottom value" for areas deeper than 50 m? Please clarify.

**AR19:** According to the report released by the Ministry of the Environment, Japan (https://www.env.go.jp/content/900530598.pdf), it explained that the original data for those stations deeper than 50 m were sampled at 50 m in the Broad Comprehensive Water Quality Survey.

In order to avoid misunderstanding the definition of the bottom layer, we will modify the related sentences in the revised manuscript.

The revised expression is:

"At each station, the data were sampled from two layers: the upper layer, located 1 m below the sea surface, and the lower layer, positioned 1 m above the sea floor for stations shallower than 50 m. For stations deeper than 50 m, the data for the lower layer were obtained at a fixed depth of 50 m (https://www.env.go.jp/content/900530598.pdf).".

**RC20:** Section 3.1: While Fig. 2 and Fig. 3 are good for showing how well the model captures the spatial signal, it is really hard to see by eye whether it also resolves the seasonal patterns. I suggest adding a few climatologies from the model compared to observations, for a few stations representative for different subareas of the model domain. This is probably sufficient in the supplement.

**AR20:** Thank you for your suggestions. The observation data from the Ministry of the Environment, Japan covers the whole SIS, and the sampling date in January, May, July, and October, representing winter, spring, summer, and autumn. Here, we show you the observed DIN and PHY concentrations averaged in each sub-region of SIS in the four seasons. Then we will calculate the simulated DIN and PHY concentration of monthly mean in each sub-region of SIS and the comparison with the observations will be provided in the Supplement Materials of the revised manuscript.

**Table A1**. Monthly mean DIN concentration (mmol m$^{-3}$) averaged from 1980 to 2018 in the SIS and its sub-regions. The value before the slash is for the upper layer, and that after the slash is for the lower layer.

|  | Jan. | May | July | Oct. |
|---|---|---|---|---|
| **Seto Inland Sea** | 4.57/4.25 | 1.71/2.42 | 1.79/3.74 | 3.56/4.00 |
| **Bungo Channel** | 4.36/4.66 | 1.04/2.06 | 1.03/3.40 | 2.96/3.64 |
| **Iyo Nada** | 2.78/2.92 | 0.67/1.78 | 0.63/2.87 | 2.11/3.28 |
| **Suo Nada** | 1.02/1.00 | 0.58/0.80 | 0.68/1.56 | 1.33/1.69 |
| **Aki Nada** | 5.17/5.33 | 1.76/2.24 | 2.21/3.75 | 4.18/4.70 |
| **Hiroshima Bay** | 2.86/3.28 | 0.81/0.93 | 0.90/1.84 | 2.23/2.84 |
| **Hiuchi Nada** | 4.47/4.65 | 0.87/1.02 | 1.23/1.78 | 2.57/2.47 |
| **Bingo Nada** | 2.89/3.19 | 1.38/1.61 | 1.64/3.37 | 2.44/2.30 |
| **Bisan Strait** | 3.19/3.19 | 1.93/2.01 | 3.73/3.67 | 8.37/7.67 |
| **Harima Nada** | 4.42/4.60 | 1.39/4.03 | 1.07/5.62 | 3.82/5.04 |
| **Osaka Bay** | 11.82/7.13 | 5.67/3.76 | 4.16/5.84 | 5.60/5.36 |
| **Kii Channel** | 7.31/6.85 | 2.65/6.34 | 2.44/7.42 | 3.53/5.00 |

**Table A2**. Monthly mean PHY concentration (mg Chla m$^{-3}$) averaged from 1980 to 2018 in the SIS and its sub-regions. The value before the slash is for the upper layer, and that is after the slash for the lower layer.

|  | Jan. | May | July | Oct. |
|---|---|---|---|---|

| | | | | |
|---|---|---|---|---|
| **Seto Inland Sea** | 1.79/1.70 | 1.45/1.16 | 2.93/1.32 | 2.23/1.85 |
| **Bungo Channel** | 0.72/0.68 | 0.93/1.76 | 1.21/0.64 | 1.12/0.76 |
| **Iyo Nada** | 0.88/0.94 | 0.48/0.28 | 0.53/0.52 | 1.62/1.18 |
| **Suo Nada** | 1.56/1.99 | 1.41/1.85 | 1.24/1.48 | 1.75/2.03 |
| **Aki Nada** | 1.17/1.30 | 0.71/0.91 | 1.31/1.27 | 2.06/1.97 |
| **Hiroshima Bay** | 1.46/1.52 | 1.07/1.13 | 1.25/1.53 | 1.87/1.98 |
| **Hiuchi Nada** | 1.28/1.54 | 0.93/1.21 | 1.05/1.72 | 1.74/2.11 |
| **Bingo Nada** | 3.18/3.46 | 1.33/2.29 | 1.62/2.59 | 2.84/3.01 |
| **Bisan Strait** | 2.38/2.65 | 2.38/2.46 | 2.68/2.75 | 2.35/2.79 |
| **Harima Nada** | 0.86/1.33 | 0.86/0.57 | 1.29/0.50 | 1.23/1.33 |
| **Osaka Bay** | 5.56/2.68 | 5.53/1.03 | 18.73/1.32 | 7.19/2.78 |
| **Kii Channel** | 0.63/0.56 | 0.32/0.23 | 1.29/0.22 | 0.77/0.37 |

In addition, we also collected observed data reported in the literatures in several sub-regions of SIS to validate the seasonal variation of DIN and PHY. The collected data are presented below and the comparison with simulated results will also be provided in the Supplement Materials of the revised manuscript.

There are observed DIN and PHY concentrations averaged from 0 to 40 m depth in Iyo Nada and Bungo Channel in 2009 (Yoshie et al., 2011). We extracted these data from the figures of Yoshie et al. (2011) and organized them in the tables below for future reference.

**Table A3**. DIN concentration (mmol m$^{-3}$) and PHY concentration (mg Chla m$^{-3}$) averaged from 0 to 40 m depth in Iyo Nada (Yoshie et al., 2011).

| | DIN | PHY |
|---|---|---|
| **Jan.** | No data | No data |
| **Feb.** | No data | No data |
| **Mar.** | No data | No data |
| **Apr.** | 0.73±0.20 | 0.97±0.10 |
| **May** | 0.70±0.07 | 0.79±+0.12 |
| **June** | 1.09±0.25 | 1.28±0.20 |
| **July** | 1.40±0.36 | 1.22±0.29 |
| **Aug.** | 1.81±0.50 | 2.12±0.80 |
| **Sept.** | 1.72±0.30 | 2.29±0.35 |

| | | |
|---|---|---|
| Oct. | 2.62±0.30 | 1.74±0.20 |
| Nov. | 4.11±0.40 | 1.32±0.21 |
| Dec. | No data | No data |

**Table A4**. DIN concentration (mmol m$^{-3}$) and PHY concentration (mg Chla m$^{-3}$) averaged from 0 to 40 m depth in Bungo Channel (Yoshie et al., 2011).

| | DIN | PHY |
|---|---|---|
| **Jan.** | No data | No data |
| **Feb.** | No data | No data |
| **Mar.** | No data | No data |
| **Apr.** | 0.75±0.26 | 0.94±0.49 |
| **May** | 1.23±0.30 | 0.51±0.20 |
| **June** | 1.39±0.45 | 1.06±0.22 |
| **July** | 2.72±0.43 | 1.08±0.35 |
| **Aug.** | 2.18±0.24 | 1.49±0.32 |
| **Sept.** | 2.34±0.00 | 1.53±0.00 |
| **Oct.** | 2.58±0.72 | 1.23±0.58 |
| **Nov.** | 3.27±0.00 | 0.90±0.00 |
| **Dec.** | No data | No data |

In Harima Nada, Nishikawa et al. (2010) described the seasonal variations in DIN concentration at the surface using monthly monitoring data obtained from April 1973 to December 2007. We organized these data in Table A5.

**Table A5.** DIN concentration (mmol m$^{-3}$) at the surface layer in Harima Nada (Nishikawa et al., 2010).

| | DIN |
|---|---|
| **Jan.** | 8.4±3.3 |
| **Feb.** | 6.1±3.1 |
| **Mar.** | 3.9±2.6 |
| **Apr.** | 4.2±2.7 |
| **May** | 4.1±2.5 |
| **June** | 3.6±2.8 |
| **July** | 4.1±4.0 |
| **Aug.** | 2.1±1.9 |
| **Sept.** | 2.6±2.6 |

| | |
|---|---|
| Oct. | 6.1±2.6 |
| Nov. | 7.5±2.8 |
| Dec. | 9.6±3.2 |

In Harima Nada, Kobayashi and Fujiwara. (2008) also reported the seasonal variations of surface and bottom DIN concentration. We organized these data in Table A6.

Table A6. DIN concentration (mmol m$^{-3}$) at the surface and bottom layers in Harima Nada (Kobayashi and Fujiwara., 2008).

| | Surface DIN | Bottom DIN |
|---|---|---|
| Jan. | 8.16 | 8.41 |
| Feb. | 8.34 | 8.20 |
| Mar. | 7.35 | 7.18 |
| Apr. | 2.91 | 4.32 |
| May | 1.94 | 4.11 |
| June | 1.32 | 5.11 |
| July | 2.71 | 7.32 |
| Aug. | 1.02 | 10.72 |
| Sept. | 1.15 | 11.12 |
| Oct. | 1.07 | 11.12 |
| Nov. | 4.65 | 7.15 |
| Dec. | 6.00 | 6.45 |

**Reference:**

Kobayashi, S. and Fujiwara, T.: Long-term variability of shelf water intrusion and its influence on hydrographic and biogeochemical properties of the Seto Inland Sea, Japan, J Oceanogr, 64, 595–603, https://doi.org/10.1007/s10872-008-0050-0, 2008.

Nishikawa, T., Hori, Y., Nagai, S., Miyahara, K., Nakamura, Y., Harada, K., Tanda, M., Manabe, T., and Tada, K.: Nutrient and Phytoplankton Dynamics in Harima-Nada, Eastern Seto Inland Sea, Japan During a 35-Year Period from 1973 to 2007, Estuaries and Coasts, 33, 417–427, https://doi.org/10.1007/s12237-009-9198-0, 2010.

Yoshie, N., Guo, X., Fujii, N., and Komorita, T.: Ecosystem and nutrient dynamics in the Seto Inland Sea, Japan, Interdisciplinary Studies on Environmental Chemistry, Modeling and Analysis of Marine Environmental Problems, 5, 39–49, 2011.

**RC21:** Line 234-238: "have already occupied most areas of the SIS": it would be better to calculate the ratio

(DIN_ocean+DIN_river+DIN_sediment)/DIN. If that is close to one everywhere in the model domain, you can estimate that your spin-up period for the tagging is completed.

**AR21:** We will add one figure to describe the ratio of (DIN_ocean+DIN_river+DIN_sediment)/DIN in the SIS from the first year to the third year of the tracking simulation in the Supplement Materials of the revised manuscript.

**RC22:** Line 373: "whose ratio is 1.4:1": The ratio between what? Subsequently more occurrences.

**AR22:** This ratio is between the horizontal export flux of biological particles (PHY+ZOO+PON) to the open ocean (187 mol N s$^{-1}$) and the vertical export flux of biological particles to the sediment (136 mol N s$^{-1}$).

187 mol N s$^{-1}$/ 136 mol N s$^{-1}$≈1.4

We will give more explanations in the revised manuscript.

The revised expression is:

"In the SIS, the horizontal export flux of biological particles (PHY+ZOO+PON) to the open ocean is 187 $mol\ s^{-1}$ (Fig. 9a) and the vertical export flux of biological particles to the sediment is 136 mol N s$^{-1}$, whose ratio is 187 mol N s$^{-1}$/136 mol N s$^{-1}$≈1.4:1.".

"For oceanic nutrients (Fig. 9b), the horizontal export of biological particles has a flux of 142 $mol\ s^{-1}$ while the vertical export has a value of 62 $mol\ s^{-1}$, whose ratio is 142 mol N s$^{-1}$/62 mol N s$^{-1}$≈2.3:1; for the riverine nutrients (Fig. 9c), the horizontal export has a flux of 23 $mol\ s^{-1}$ while the vertical export has a value of 27 $mol\ s^{-1}$, whose ratio is 23 mol N s$^{-1}$/27 mol N s$^{-1}$≈0.85:1; for the benthic nutrients (Fig. 9d), the horizontal export has a flux of 22 $mol\ s^{-1}$ while the vertical export has a value of 47 $mol\ s^{-1}$, whose ratio is 22 mol N s$^{-1}$/47 mol N s$^{-1}$≈0.48:1.".

**RC23:** Line 454: "the management can also be applied to the sediments": It is very unclear how you would "manage" sedimentary nutrient release, you cannot easily modify it. If this is a serious option please give more details, e.g. will you add substances to capture some of the escaping nutrients?

**AR23:** Thank you for your comment. We intend to raise awareness about the sediment release in the SIS. As you stated, the expression "the management can also be applied to the sediments" is not appropriate. We modify this sentence in the revised manuscript.

The revised expression is:

"it needed to pay more attention on the sediments".

---

## Author Response (AR2)

**Response to Reviewer #1**

**Always given as follows:**

**Referee comment: RC;** Author response: AR; Changes to the manuscript

**General comments**

This is a numerical model study to identify the contribution of three sources of nitrogen, which is from the land, from the open sea, and from the seafloor to DIN in a semi-enclosed coastal sea (Seto Inland Sea: SIS). Although this information is important for the prevention of eutrophication in the sea, the calculation method has some problems as shown below.

Thank you for your valuable comments on this study. We have carefully considered your comments regarding the calculation method and would like to address them in this response note.

**Specific comments**

**RC1:** Why do you include seafloor sources as a nitrogen source in addition to the open sea and terrestrial sources?

Unlike the case of phosphorus, nitrogen leached from the seafloor is the result of mineralization of "new" sediments, so nitrogen originating from the seafloor may be included in the open sea nitrogen and land nitrogen.

**AR1:** Thank you for your suggestion. We agree with your idea that the nitrogen leached from the seafloor is the result of the mineralization of "new" sediments. The nutrients contained in "new" sediments have an initial origin from either the rivers or the open ocean. The reason we treated it as a source is that many scientists always argue about the contribution of sediment to the nutrient in the water column. Therefore, this is just a practice way to define the nutrient source.

As we consider the sediment source of nutrients, we also realize that it has different processes from those from rivers and the open ocean. First, the processes by which particulate organic nitrogen (PON) settles onto the sediment, decomposes into dissolved inorganic nitrogen (DIN), and returns to the water column are different from the directly discharged nutrients of riverine or oceanic DIN, as well as the remineralization of PON in the water column. The sediment-released nutrients are suggested to be particularly important in shallow waters (Radtke et al., 2019). Second, there is a time lag between the deposition and mineralization of PON, either short-term or long-term (Soetaert et al., 2000). This means a large uncertainty in the proportion of nutrients released from sediment as compared to those supplied from rivers and the open ocean. Third, it is more difficult to control the nutrients released from the sediment than those supplied by rivers. Therefore, identifying the areas where the sediment-released nutrients dominate is helpful to the effective control or regulation of the riverine nutrients.

In conclusion, it is still necessary to track the sediment-released nutrients as a separate source. To address your comment, we will mention that the nutrients leached from the seafloor have an origin from either rivers or the open ocean in Section 2.3. In addition, based on the proportion of PON flux settled on the sediment surface originating from the open ocean and rivers, we will give a quantitative estimation of the ratio of

riverine and oceanic nutrients in the sediment-released nutrients in Section 4.2.

**Reference:**

Radtke, H., Lipka, M., Bunke, D., Morys, C., Woelfel, J., Cahill, B., Böttcher, M. E., Forster, S., Leipe, T., Rehder, G., and Neumann, T.: Ecological ReGional Ocean Model with vertically resolved sediments (ERGOM SED 1.0): coupling benthic and pelagic biogeochemistry of the south-western Baltic Sea, Geosci. Model Dev., 12, 275–320, https://doi.org/10.5194/gmd-12-275-2019, 2019.

Soetaert, K., Middelburg, J. J., Herman, P. M., and Buis, K.: On the coupling of benthic and pelagic biogeochemical models, Earth-Science Reviews, 51, 173–201, 2000.

The sentences added to the revised manuscript are:

Added in Section 2.3:

"It needs to note that the nutrients in the sediment are initially from either the land or the open ocean and the sediment acts a temporary storage or a permanent sink of these nutrients. In this study, however, we treat the sediment as the third source to track. This is because the sediment-released nutrients are gaining more attention and are particularly important in shallow waters (Radtke et al., 2019)."

Added in Section 4.2:

"If the proportion of PON flux settled on the sediment surface produced by the oceanic and riverine nutrients were taken as the proportion of DIN flux released from the sediment, among the 86 $mol\ s^{-1}$ of DIN from the sediment, 37 $mol\ s^{-1}$ has an origin in the open ocean and 49 $mol\ s^{-1}$ from rivers."

**RC2:** Why is dissolved organic nitrogen not included in the calculation?

Dissolved organic nitrogen, which accounts for about 90% of the total nitrogen in SIS, is not included in the calculation. The Ministry of the Environment's total load reduction for SIS is also based on total nitrogen in its calculations.

**AR2:** Thank you for your comment. Each biogeochemical model is designed for a different purpose, and the number and type of variables and biogeochemical processes are selected depending on the purpose (Fennel et al., 2022). The introduction of more variables and processes will introduce more uncertainties and calculation costs. In addition to expressing biogeochemical processes explicitly, expressing them implicitly is a substitute. The purpose of this study is to evaluate the inventories of nutrients and phytoplankton originating from different sources in the SIS. The direct transformations between them are of the most concern. Therefore, the widely used NPZD model including DIN, PHY, ZOO, and PON, is sufficient for the research purpose. For example, to estimate the nitrogen fluxes in the shelf area of the Middle Atlantic Bight, Fennel et al. (2006) also used an NPZD model to represent the nitrogen processes in the water column.

Although the research about the dissolved organic nitrogen (DON) in SIS was not often reported and the dataset of DON was also little, we still collected some results. In Hiuchi Nada and Iyo Nada, which are two sub-regions of SIS, Kumamoto et al. (1994) estimated the DON concentration at two sampling stations in late spring. They reported that DON concentration was superior to DIN concentration and nearly constant in the vertical and showed little temporal variation in Iyo Nada. By subtracting DIN from TN, Asahi et al. (2019) obtained the DON concentration at the surface layer of SIS in the 1990s and 2000s. They reported that DON

concentration was higher than DIN and was relatively constant. Because we are interested in the materials that were dynamically changed, we did not include the little changed DON in our model. In other words, DON is like a background for TN but only DIN has a strong relationship with the low-trophic ecosystem. After reading your comment, however, we acknowledge that the incorporation of DON can be considered for some specific questions if there is a sufficient amount of observational DON to support model construction.

**Reference:**

Asahi T., Abo K., Abe K., and Tada K.: Comparison of Dissolved Inorganic and Organic Nitrogen between the sand s in the Seto Inland Sea of Japan, Bulletin on Coastal Oceanography, 56, 123–131, 2019.

Kumamoto, Y., Tsubota, H., and Fujiwara, K.: Temporal Variation of Dissolved Organic Nitrogen and Phosphorus, Environmental Science, 7, 1–12, https://doi.org/10.11353/sesj1988.7.1, 1994.

**RC3:** Are the DIN boundary conditions used in the numerical model reasonable?

The boundary condition is that the open sea origin DIN is zero at the seafloor and landward.

If there are no biochemical processes in SIS, no DIN supply from land or seafloor, only physical diffusion, then the DIN concentration in SIS is equal to the open boundary DIN (DIN from the open sea) and SIS is filled with DIN from the open sea. In other words, DIN = 0 does not occur on the seafloor surface or the landward shore.

**AR3:** Yes, the DIN boundary conditions used in the numerical model and the tracking cases are reasonable. From the mathematical perspective, the numerical model is represented as a system of coupled partial differential equations, which are the same in the calculations for three types of nutrients and their sum. The tracking method we used in this study is a linear decomposition of boundary conditions used for the partial differential equations. For the boundary condition of the tracking open ocean case, we specified the open ocean origin DIN having a zero flux at the land boundary and seafloor. This is because the open ocean origin DIN is not released into the SIS from the land or seafloor.

The case you said that SIS is filled with DIN from the open sea is the solution of a diffusion equation with zero flux from land and a fixed concentration at the open ocean side. We actually used the same boundary conditions as the case you said. In our results, the concentration of open ocean-origin DIN is also not zero at the landward shore or on the seafloor surface (Fig. 6b and Fig. 6e).

**Technical correction**

RC4: It should be noted that the land load in this report is an underestimate.

The terrestrial nitrogen load for SIS is published every five years by the Ministry of the Environment of Japan (MEJ). It is necessary to state the values of the terrestrial load by MEJ and the terrestrial load in this report.

In SIS, which experienced eutrophication in the 1970s, the majority of domestic and industrial wastewater is treated at treatment facilities on the waterfront and discharged directly into the sea in recent years. Therefore, there is a large difference between the DIN flow via rivers and the total nitrogen flow actually

entering the sea (especially in the eastern Seto Inland Sea).

In SIS, river discharge is significantly lower in winter, resulting in large seasonal variations in DIN flow from rivers, whereas there is little seasonal variation in DIN flow from domestic and industrial sources. This affects the seasonal variation of DIN concentration in the SIS.

**AR4:** Thank you for your suggestions. We have recognized that the DIN load from the land was underestimated in our study, and we mentioned it on Line 115 to Line 117 of the original manuscript. In the revised manuscript, we will appropriately increase the DIN load from the land using the total nitrogen (TN) load for SIS published every five years by the Ministry of the Environment of Japan, and we will give a quantitative estimate of how large this underestimation in the DIN load from land is. In this response letter, we first report the related information we have collected.

TN loads from land to the SIS were estimated by the Ministry of the Environment, Japan every five years from 1979 based on the unit load method in the catchment area of SIS (Abo and Yamamoto, 2019; Timita et al., 2016). From 1979 to 2014, the average TN load from land to the SIS was 471 mol s$^{-1}$. Yanagi and Ishii (2004) indicated that the TN loads estimated by the unit load method did not reproduce the inflow of TN to the coastal sea since some parts of TN loads remained on land. Yamamoto et al. (1996) recommended the use of river flow rate to calculate the actual inflow TN load into the SIS and reported that the TN load calculated using this method was about 48% of that measured by the unit load method. Based on this value presented by Yamamoto et al. (1996), the average inflow TN load to the SIS was 226 mol s$^{-1}$, which flowed into the SIS from the land through the 21 first-order rivers and about 640 other small rivers. We included 21 first-order rivers and 45 small rivers in our study. In addition, the compounds of TN loads from land are not clear.

The proportion of DIN concentration in TN concentration at the first-order rivers of SIS is about 77%, which was estimated by the nutrient data from the Ministry of Land, Infrastructure, Transport and Tourism, Japan (http://www1.river.go.jp/). If we apply this proportion to the TN load from land, the DIN load from land is 174 mol s$^{-1}$, which is 2.7 times higher than the DIN load from rivers (64 mol s$^{-1}$) estimated by our study. In order to consider the DIN load from land in revision, a new series of experiments is conducted by increasing the DIN load from the rivers to 3 times of the original value (64 mol s$^{-1}$) to represent the DIN load from land. The results of new experiments have been described in the revised manuscript.

**Reference:**

Abo, K. and Yamamoto, T.: Oligotrophication and its measures in the Seto Inland Sea, Japan, Bulletin of Japan Fisheries Research and Education Agency, 49, 21–26, 2019.

Tomita, A., Nakura, Y., and Ishikawa, T.: New direction for environmental water management, Marine Pollution Bulletin, 102, 323–328, https://doi.org/10.1016/j.marpolbul.2015.07.068, 2016.

Yamamoto, T., Kitamura, T., and Matsuda, O.: Riverine inputs of fresh water, total nitrogen and total phosphorus into the Seto Inland Sea, Journal of the Faculty of Applied Biological Science, Hiroshima University, 35, 81–104, 1996.

Yanagi, T. and Ishii, D.: Open Ocean Originated Phosphorus and Nitrogen in the Seto Inland Sea, Japan, J Oceanogr, 60, 1001–1005, https://doi.org/10.1007/s10872-005-0008-4, 2004.

The sentences added to the revised manuscript are:

Added in Section 2.2:

"The Ministry of the Environment, Japan estimated the total nitrogen (TN) loads of an average value of 471 $mol\ s^{-1}$ from land to the SIS from 1979 to 2014 based on the unit method (Abo and Yamamoto, 2019; Tomita et al., 2016; Yamamoto, 2003). Yanagi and Ishii (2004) indicated that the TN load estimated by this unit load method did not represent the input of TN into the coastal sea since some parts of the TN load remained on land. Yamamoto et al. (1996) recommended the use of river flow rate to calculate the actual inflow TN load into the SIS and reported that the actual TN load into the SIS calculated using this method was about 48% of that given by the unit load method. Based on this proportion, the average input of TN load to the SIS was 471×48%=226 $mol\ s^{-1}$.

The proportion of DIN concentration in TN concentration at the first-order rivers of SIS is about 77%, which was estimated by the nutrient data from MILT. Applying this proportion to the TN load from land, the DIN load from land is 226×77%=174 $mol\ s^{-1}$. In our study, we used daily river discharge and monthly nutrient concentration from MILT averaged over the period from the 1990s to the 2010s to estimate the DIN load from rivers into the SIS. The annual mean of DIN loads from rivers is 63.85 $mol\ s^{-1}$, which was 37% of the expected DIN load from land into the SIS. To consider the DIN load from land as realistic as possible, the DIN load from rivers was increased to 3 times of the original value to represent the DIN load from land. Consequently, the annual mean of DIN loads from land becomes about 192 $mol\ s^{-1}$. There is a clear spatial variation in DIN loads from land, which shows high DIN loads in the eastern part such as Osaka Bay and Harima Nada (Fig. S2). The seasonal variation in DIN loads, showing high values in July and September and low value in January, is primarily controlled by the seasonal variation in river discharge. We did not include particle nitrogen input from land."

**RC5:** Section 3.2. It is important to indicate the time required for the numerical model to become stationary; the DIN flow path during the set-up period is not the flow path when the model becomes stationary.

**AR5:** Thank you for your suggestions. We indicated the time required for the numerical model to become stationary in Section 2.2. In Lines 150 to 151 of the original manuscript: "The hydrodynamic-biogeochemical model was initiated on the first day of January and stabilized from the third year onwards. Therefore, the simulation results of the third year were used to analyze the seasonal variations of DIN and PHY.". For the tracking case, in Line 171 of the original manuscript: "We initiated the tracking technique from the first day of the fourth year of the hydrodynamic-biogeochemical model, …" and at Line178 to 179 of the original manuscript: "After a spin-up of three years, the annual cycle of each source of nutrients and related particles became stationary.". In order to make readers clear about the time required for the numerical model to become stationary, we added one figure to describe the ratio of (DIN_ocean+DIN_river+DIN_sediment)/DIN in the SIS from the first year to the third year of the tracking simulation in the revised Supplement Materials.

The purpose of Section 3.2 (new Section 3.3) is to exhibit the pathway of oceanic, riverine, and benthic DIN gradually occupying the SIS from the initial state. After one year of calculation, DIN concentrations from

the open ocean, rivers, and sediment have already occupied most areas of the SIS. Therefore, we presented the results of the first year of tracking cases in Fig. 4 to depict this pathway.

Added in Supplement Materials:

[Figure]

**Figure S8. (a-r)** The monthly mean ratio of water column averaged $(DIN_{ocean} + DIN_{river} + DIN_{sediment})/DIN$ over the first three years. **(s)** The daily mean ratio of volume averaged $(DIN_{ocean} + DIN_{river} + DIN_{sediment})/DIN$ over the first three years.

**RC6:** Actual measurements of the amount of nitrogen and phosphorus entering SIS from the open sea were made by several organizations in the 1980s to 2000s, and it has been shown that the amount of nitrogen entering from the open sea is equivalent to the amount of land-based load during the summer months. It is desirable to cite these papers.

**AR6:** Thank you for your suggestions. In revision, we collected the related papers and present them in this response note. We will cite these papers in Section 4.2 of the revised manuscript.

Fujiwara et al. (1997) reported 140 mol s$^{-1}$ of DIN through Kii Channel from the open ocean based on observations over 2 days in August of 1985. Kasai et al. (2001) estimated the net DIN transport through Kii Channel into SIS in August of 1996 was 111 mol s$^{-1}$ which was similar to that of Fujiwara et al. (1997).

By subtracting the amount of DIN load buried in the sediment from the amount of DIN load from land, Fujiwara et al. (2006) reported that 50 mol s$^{-1}$ of TN was transported from the open ocean to the SIS.

**Reference:**

Fujiwara, T., Uno, N., Tada, M., Nakatsuji, K., Kasai, A., and Sakamoto, W.: Inflow of nitrogen and phosphorus from the ocean into the Seto Inland Sea, Proc. Coastal Engineering (JSCE), https://doi.org/10.2208/proce1989.44.1061, 1997.

Fujiwara, T., Kobayashi, S., Kunii, M., and Uno, N.: Nitrogen and phosphorus in Seto Inland Sea: Their origin, budget and variability, Bull Coast Oceanogr, 43, 129–136, 2006.

Kasai, A., Fujiwara, T., and Tada, M.: Ocean Structure and Nutrient Transport in the Kii Channel, Proceedings of Coastal Engineering, 48, 436–440, https://doi.org/10.2208/proce1989.48.436, 2001.

Added in Section 4.2:

"Fujiwara et al. (1997) reported about 168 $mol\ s^{-1}$ of DIN through Kii Channel from the open ocean in August of 1985. Kasai et al. (2001) estimated the net DIN transport through Kii Channel into SIS in August of 1996 was 111 $mol\ s^{-1}$ which was similar to that of Fujiwara et al. (1997). We estimated about 150 $mol\ s^{-1}$ of DIN was from the open ocean to the SIS through Kii Channel in August."

"Fujiwara et al. (2006) also reported that the net input of TN from the open ocean was 50 $mol\ s^{-1}$, which was buried in the sediment. In our study, the TN originating from the open ocean has a net input of 61 $mol\ s^{-1}$, all of which were buried in the sediment."

**RC7:** Line 114: It should be noted that the seasonal variation of the nitrogen load from rivers is due to the seasonal variation of the river flow. Unlike Europe, the SIS receives a little precipitation in winter.

**AR7:** Thank you for your suggestions. We will note this information in the revised manuscript.

The revised expression is:

"The seasonal variation in DIN loads, with high loads in July and September and low loads in January, is primarily controlled by the seasonal variation in river discharge."

**Response to Reviewer #2**

**Always given as follows:**

**Referee comment: RC; Author response: AR; Changes to the manuscript**

**General comments**

**RC1:** Information about the hydrodynamics of the SIS is missing, specifically the main currents should be described and compared between your model and e.g., literature references, since it is essential that the advective transport is realistically captured, which is not obvious from validating the DIN and DIP concentrations alone. Also comparing salinity to observations might help to check whether the mixing ratio between riverine and oceanic water masses is realistically captured in the model.

**AR1:** Thank you for your suggestions. The hydrodynamic model used in this study was the same as that in Chang et al. (2009) and Zhu et al. (2019). They have finished the general comparisons with observations (residual current pattern, monthly water temperature, and monthly salinity) for the hydrodynamic model, which confirmed that this model can generally reproduce the major hydrodynamic characteristics of the Seto Inland Sea (SIS). In addition, this model has been used to study the formation of cold bottom water and some related processes (Yu et al., 2016; Yu and Guo, 2018) as well as to calculate the water age of river water (Wang et al., 2018). Therefore, we only cited Chang et al. (2009) and Zhu et al. (2019) in the original manuscript but did not give a detailed description, which is shown as follows:

"Chang et al. (2009) compared the simulated surface residual current of this SIS hydrodynamic model with the observations. It showed that the summer and winter circulation patterns were reproduced. Significant cyclonic and anticyclonic eddies were developed near the entrance and inner part of Suo Nada, respectively, both in the simulated and observed results. In addition, the model also captured the southward current flowing to the western Bungo Channel and the southwestward current in the northern Iyo Nada, which were also evident in the observations. The model also well reproduced the observed circulation features in the Harima Nada."

"Zhu et al. (2019) compared the simulated temperature and salinity of SIS in February (winter) and July (summer) with the observations. They reported that the warm and saline waters flowed into the SIS through the Bungo Channel and Kii Channel in winter, which was consistent with the observations. For the vertical distributions in winter, both the simulated and observed results showed that the water column was well mixed throughout the whole SIS. In summer, both temperature and salinity exhibited a well-mixed pattern around the straits and a well-stratified pattern in the broad basins. Low salinity existed in Osaka Bay, forming a front structure with high salinity water in the Kii Channel."

In the revised manuscript, based on your suggestions, we described the main current field of SIS and compare the model results with the literature in Section 3.1. For the salinity distribution, we have obtained a long-term monthly observation dataset carried out by the prefectural fishery research centers around the SIS. We also compared the simulated salinity results with observations in Section 3.1 of the revised manuscript.

**Reference:**

Chang, P.-H., Guo, X., and Takeoka, H.: A numerical study of the seasonal circulation in the Seto Inland Sea, Japan, J. Oceanogr., 65, 721–736, https://doi.org/10.1007/s10872-009-0062-4, 2009.

Wang, H., Guo, X., and Liu, Z.: The age of Yodo River water in the Seto Inland Sea, Journal of Marine Systems, 191, 24–37, https://doi.org/10.1016/j.jmarsys.2018.12.001, 2019.

Yu, X. and Guo, X.: Intensification of water temperature increase inside the bottom cold water by horizontal heat transport, Continental Shelf Research, 165, 26–36, https://doi.org/10.1016/j.csr.2018.06.006, 2018.

Yu, X., Guo, X., and Takeoka, H.: Fortnightly Variation in the Bottom Thermal Front and Associated Circulation in a Semienclosed Sea, Journal of Physical Oceanography, 46, 159–177, https://doi.org/10.1175/JPO-D-15-0071.1, 2016.

Zhu, J., Guo, X., Shi, J., and Gao, H.: Dilution characteristics of riverine input contaminants in the Seto Inland Sea, Mar. Pollut. Bull., 141, 91–103, https://doi.org/10.1016/j.marpolbul.2019.02.029, 2019.

Added in Section 2.4:

"Chang et al. (2009) and Zhu et al. (2019) have conducted a comprehensive comparison between the simulated residual current pattern, monthly water temperature, and monthly salinity and the corresponding observations. We provided a concise summary of their main results in Section 3.1. Additionally, to check whether the mixing ratio between the riverine and oceanic water masses was realistically captured by the model, we compared the simulated sea surface salinity (SSS) with the observations. The observed sea surface salinity was provided by the Fisheries Research Center, Ehime Prefectural Institute of Agriculture, Forestry and Fisheries, Japan and these data was averaged from 1980 to 2001 to derive the climatological distribution."

Added in Section 3.1:

**3.1 Seasonal and spatial variations of sea surface salinity**

Chang et al. (2009) demonstrated that the summer and winter circulation patterns were reproduced by this hydrodynamic model. Significant cyclonic and anticyclonic eddies were developed near the entrance and inner part of Suo Nada, respectively. Moreover, the model also captured the southward current flowing into the western Bungo Channel and the southwestward current in the northern Iyo Nada. Zhu et al. (2019) reported that the warm and saline waters flowed into the SIS through the Bungo Channel and Kii Channel in winter. For the vertical distributions in winter, both the simulated and observed results showed that the water column was well mixed throughout the whole SIS. In summer, both temperature and salinity exhibited a well-mixed pattern around the straits and a well-stratified pattern in the broad basins.

We conducted a comparison between the simulated SSS and the observed one in January (winter), May (spring), July (summer), and October (autumn) (Fig. 2). Throughout the year, the water with a salinity of 33-34 covered the surface layer of Bungo Channel and Kii Channel, among which the SSS was larger in Bungo Channel than in Kii Channel. In January, the SSS was consistently around 33-34 in most areas of SIS. In May, the SSS decreased in the inshore areas, particularly in the eastern part. In July, the SSS in most inshore areas dropped below 32. In October, the water of lower salinity was refined to the nearshore areas. The

hydrodynamic model of this study effectively reproduced the observed SSS distribution in the SIS, indicating a realistic capture of the mixing ratio between riverine and oceanic water masses.

[Figure]

**Figure 2.** Monthly mean **(a-d)** observed and **(e-h)** simulated sea surface salinity.

**RC2**: You state that your nitrogen loads are smaller than previously reported values, as you neglect industrial and land-based sources as well as particulate forms of riverine nitrogen. It seems you also ignore atmospheric deposition? Since the main result of the paper, which is the oceanic fraction of the nutrients in the ecosystem, will be strongly dependent on the terrestrial loads you put in, please give a quantitative estimate on how large this uncertainty/error in the loads is.

**AR2:** Thank you for your comment.

**First question:** Yanagi (1997) considered the deposited total nitrogen (TN) load in rainwater when estimating the nitrogen budget in SIS. In his estimation, the net TN load of atmospheric deposition was 8% of the land input. In a coastal area of SIS during the spring of 2015, the dry deposition fluxes of particulate $NH_4$ and $NO_3$ were $2.3 \times 10^{-7}$ mol m$^{-2}$ s$^{-1}$ and $5.5 \times 10^{-7}$ mol m$^{-2}$ s$^{-1}$, respectively (Nakamura et al., 2020). These atmospheric aerosols were measured on a rooftop at Kagawa College, Kagawa Prefecture, Japan and may not fully represent the atmospheric aerosols for the whole SIS. As we applied these two values to the whole SIS whose area is 23,203 km$^2$, we obtained the dry deposition fluxes of particulate $NH_4$ and $NO_3$ for

the SIS were 5.4 mol N s$^{-1}$ and 12.7 mol N s$^{-1}$, respectively. They are lower than the nitrogen input from rivers, the open ocean, and sediment (64 mol N s$^{-1}$, 174 mol N s$^{-1}$, 86 mol N s$^{-1}$). This is the result of spring and there is no study for other seasons. Considering these uncertainties, we did not include the atmospheric deposition for the SIS.

**Second question:** TN load from land to the SIS was estimated by the Ministry of the Environment, Japan every five years from 1979 based on the unit load method in the catchment area of SIS (Abo and Yamamoto, 2019; Timita et al., 2016). From 1979 to 2014, the average TN load from land to the SIS was 471 mol N s$^{-1}$. Yanagi and Ishii (2004) indicated that the TN load estimated by the unit load method did not reproduce the inflow of TN to the coastal sea since some parts of the TN load remained on land. Yamamoto et al. (1996) recommended the use of river flow rate to calculate the actual inflow TN load to the SIS and reported that the TN load calculated using this method was about 48% of that measured by the unit load method. Based on this value presented by Yamamoto et al. (1996), the average inflow TN load to the SIS was 226 mol N s$^{-1}$, which flowed into the SIS from the land through the 21 first-order rivers and about 640 other small rivers. We included 21 first rivers and 45 small rivers in our study. In addition, the compounds of TN load from land are not clear. The proportion of dissolved inorganic nitrogen (DIN) concentration in TN concentration at the first-order rivers of SIS is about 77%, which was estimated by the nutrient data from the Ministry of Land, Infrastructure, Transport and Tourism, Japan (http://www1.river.go.jp/). If we apply this value to the TN load from land, the DIN load from land is 174 mol N s$^{-1}$, which is 2.7 times higher than the DIN load from rivers (64 mol N s$^{-1}$) estimated by our study. To consider the DIN load from land as realistic as possible, a new series of experiments is conducted by increasing the DIN load from the rivers to 3 times of its original value (64 mol N s$^{-1}$) to represent the DIN load from land. The results of new experiments have been described in the revised manuscript.

**Reference:**

Abo, K. and Yamamoto, T.: Oligotrophication and its measures in the Seto Inland Sea, Japan, Bulletin of Japan Fisheries Research and Education Agency, 49, 21–26, 2019.

Nakamura, T., Narita, Y., Kanazawa, K., and Uematsu, M.: Organic Nitrogen of Atmospheric Aerosols in the Coastal Area of Seto Inland Sea, Aerosol Air Qual. Res., 20, 1016–1025, https://doi.org/10.4209/aaqr.2019.12.0658, 2020.

Tomita, A., Nakura, Y., and Ishikawa, T.: New direction for environmental water management, Marine Pollution Bulletin, 102, 323–328, https://doi.org/10.1016/j.marpolbul.2015.07.068, 2016.

Yamamoto, T., Kitamura, T., and Matsuda, O.: Riverine inputs of fresh water, total nitrogen and total phosphorus into the Seto Inland Sea, Journal of the Faculty of Applied Biological Science, Hiroshima University, 35, 81–104, 1996.

Yanagi, T.: Budgets of fresh water, nitrogen and phosphorus in the Seto Inland Sea, Umi-no-Kenkyu, 6, 157–161, 1997.

Yanagi, T. and Ishii, D.: Open Ocean Originated Phosphorus and Nitrogen in the Seto Inland Sea, Japan, J Oceanogr, 60, 1001–1005, https://doi.org/10.1007/s10872-005-0008-4, 2004.

The sentences added to the revised manuscript are:

Section 2.2

"We disregarded the DIN loads from the atmospheric deposition in the model. This is because limited studies have been conducted to observe the atmospheric nitrogen deposition flux throughout the SIS, and the reported data covered only a small area and one season (Nakamura et al., 2020). Furthermore, an estimation of the nitrogen budget in SIS reported that the nitrogen loads from atmospheric deposition were significantly smaller than that from the land through rivers (Yanagi, 1997). As we present later, because of its small contribution to the entire nutrient loads into the SIS, the model without atmospheric nitrogen deposition flux can give reasonable results for the distribution of nutrients and phytoplankton as compared to the observations."

"The Ministry of the Environment, Japan estimated the total nitrogen (TN) loads of an average value of 471 $mol\ s^{-1}$ from land to the SIS from 1979 to 2014 based on the unit method (Abo and Yamamoto, 2019; Tomita et al., 2016; Yamamoto, 2003). Yanagi and Ishii (2004) indicated that the TN load estimated by this unit load method did not represent the input of TN into the coastal sea since some parts of the TN load remained on land. Yamamoto et al. (1996) recommended the use of river flow rate to calculate the actual inflow TN load into the SIS and reported that the actual TN load into the SIS calculated using this method was about 48% of that given by the unit load method. Based on this proportion, the average input of TN load to the SIS was 471×48%=226 $mol\ s^{-1}$.

The proportion of DIN concentration in TN concentration at the first-order rivers of SIS is about 77%, which was estimated by the nutrient data from MILT. Applying this proportion to the TN load from land, the DIN load from land is 226×77%=174 $mol\ s^{-1}$. In our study, we used daily river discharge and monthly nutrient concentration from MILT averaged over the period from the 1990s to the 2010s to estimate the DIN load from rivers into the SIS. The annual mean of DIN loads from rivers is 63.85 $mol\ s^{-1}$, which was 37% of the expected DIN load from land into the SIS. To consider the DIN load from land as realistic as possible, the DIN load from rivers was increased to 3 times of the original value to represent the DIN load from land. Consequently, the annual mean of DIN loads from land becomes about 192 $mol\ s^{-1}$. There is a clear spatial variation in DIN loads from land, which shows high DIN loads in the eastern part such as Osaka Bay and Harima Nada (Fig. S2). The seasonal variation in DIN loads, showing high values in July and September and low value in January, is primarily controlled by the seasonal variation in river discharge. We did not include particle nitrogen input from land."

**RC3:** Sediment DIN flux: Your sediment model is very simplistic and maybe a bit too simplistic for your application. You assume constant DIN fluxes from the sediments in a study where you state that your goal is to understand the temporal dynamics of eutrophication. You ignore a positive feedback loop in which enhanced nutrient loads lead to more settling PON, to higher reactive TN concentrations in the surface sediment and subsequently to higher DIN release from the sediments. Please at least discuss the potential implications of this strong simplification in your discussion section. This is especially critical since sedimentwater DIN fluxes are not easily observable. They tend to show substantial small-scale variation depending on e.g. the presence of bioturbating or bioirrigating macrofauna. Please give more information on what the uncertainty of the benthic flux estimates is.

**AR3:** Thank you for your suggestions. In our study, the DIN flux from the sediment was calculated by the surface sediment TN concentration and bottom temperature based on an empirical function (Tada et al., 2018). This empirical function is based on the measured DIN flux in the laboratory using the sediment collected in many stations in the SIS (Tada et al., 2018). The reason we used it is because it reflects the actual situation in the SIS.

As we applied this formula in this study, we used the mean sediment surface TN concentration averaged from the observation data in the past 40 years, which reflected only an average state of surface TN concentration over this period and therefore ignored its long-term trend. Because the surface sediment TN concentration used in the formula is independent of the particle flux from the water column, it has not the feedback dynamic you mentioned. In fact, the only temporal variation in the DIN flux from the sediment in this study was induced by the annual variations of the bottom temperature derived from the hydrodynamic model.

Because we included the process of resuspension of particles from the sediment surface and its decomposition in the water column, we think that the short-term effects of PON settled to the sediment surface have been treated as a part of the nitrogen cycle processes in the water column. On the other hand, we also feel that it is not reasonable to treat the DIN flux from such short-term effects as a source of nutrients. In other words, our benthic nutrient flux reflects only the long-term one whose timescale is close to one year.

To make these points a little clear, we will add some sentences in the revised manuscript.

**Reference:**

Tada K., Nakajima M., Yamaguchi H., Asahi T., and Ichimi K.: The Nutrient Dynamics and Bottom Sediment in Coastal Water, Bull. Coast. Oceanogr., 55, 113–124, https://doi.org/10.32142/engankaiyo.55.2_113, 2018.

The revised expression is:

"It should be noted that the calculation of DIN flux released from the sediment is somewhat simple in our model. Based on an empirical function obtained from experiments (Tada et al., 2018), we used the annual mean TN concentration and bottom temperature to calculate the sediment DIN flux. This means that we did not consider the instant effect of PON settled to the surface sediment, which can increase the reactive TN concentration in the surface sediment and subsequently higher DIN flux released from the sediment. In fact, it is difficult to treat such short-term responses of benthic DIN flux to the settled PON (Soetaert et al., 2000) as a source of nutrients because they can be a part of the nutrient cycle within the water column."

**RC4:** You consider sedimentary DIN as a "source". Actually, sediments are not a source for nutrients, but just a temporary storage or a permanent sink. The nutrients stored in the sediment are originally mostly from riverine or oceanic origin. Even if this may be somehow clear for most readers, I think it is still worth

mentioning.

**AR4:** Thank you for your suggestions. We agree with your view. We will mention this information in Section 2.3. In addition, based on the proportion of PON flux settled on the sediment surface originating from the open ocean and rivers, we gave a quantitative estimation of the ratio of riverine and oceanic nutrients in the sediment-released nutrients in Section 4.2.

The sentences added to the revised manuscript are:

Added in Section 2.3:

"It needs to note that the nutrients in the sediment are initially from either the land or the open ocean and the sediment acts a temporary storage or a permanent sink of these nutrients. In this study, we treat the sediment as the third source to track. This is because the sediment-released nutrients are gaining more attention and are particularly important in shallow waters (Radtke et al., 2019)."

Added in Section 4.2:

"If the proportion of PON flux settled on the sediment surface produced by the oceanic and riverine nutrients were taken as the proportion of DIN flux released from the sediment, among the 86 $mol\ s^{-1}$ of DIN from the sediment, 37 $mol\ s^{-1}$ has an origin in the open ocean and 49 $mol\ s^{-1}$ from rivers."

**RC5:** Another point maybe worth discussing is that the "oceanic" DIN can be of riverine origin, just added to the Japanese coastal waters from rivers outside the SIS. Or is it the "open" Pacific Ocean signal that is really controlling the conditions at the borders of your model domain?

**AR5:** Thank you for your comment. First, the DIN concentration specified at the open boundaries of our model domain was derived from the relationship between the observed water temperature and DIN concentration south of the open boundaries, which was provided in the Supplement Materials. Their strong correlation at the range of lower temperatures reflects the inherent nature of water temperature and DIN concentration of the Kuroshio subsurface water. At the range of higher temperatures, the DIN concentration is low, which reflects the nutrient-poor Kuroshio surface water.

Second, some first-order rivers are flowing into the coastal waters of Kyushu, west of Japan. In principle, these river waters can pass the Bungo Channel and Kii Channel. However, as these waters reach the areas outside the SIS, they have been largely diluted by the Kuroshio. As we know, the river discharge is at an order of several hundreds of $m^3\ s^{-1}$ while the Kuroshio has a volume transport of several tens of $10^6\ m^3\ s^{-1}$. Furthermore, it needs more than one month for these waters to reach the areas outside the SIS and therefore most of the nutrients from these rivers have been used by the phytoplankton in the pathway.

For the above reasons, we concluded that the open boundary conditions really reflect the signals of the open ocean (Pacific Ocean). To make it clear to the readers, we gave more information in Section 2.2 of the revised manuscript.

Added in Section 2.2:

"Their strong correlation at the range of low temperatures reflects the inherent nature of water temperature and DIN concentration of the Kuroshio subsurface water. At the range of high temperatures, the DIN

concentration is low, which reflects the nutrient-poor Kuroshio surface water."

**RC6:** Please give some references why it is reasonable to exclude dinitrogen fixation as a relevant N source in the SIS and neglect it in the model. (in other coastal seas it is a majour source)

**AR6:** Thank you for your comment. There are several studies about dinitrogen fixation in the SIS. In Osaka Bay, which is a severely polluted sub-region of eastern SIS, Hashimoto et al. (2016) reported a nitrogen fixation of 0.0011 mol N s$^{-1}$ using the nitrogen fixation rate and cell abundance of unicellular diazotrophic cyanobacteria. This value was much lower than the nitrogen input of rivers into Osaka Bay (~42 mol N s$^{-1}$, Fig. S2). Lee et al. (1996) reported that there was no nitrogen fixation observed in Hiroshima Bay. Based on Lee et al. (1996), Yamamoto et al. (2008) assumed no nitrogen fixation in the whole SIS when estimating the nitrogen budget for the SIS. According to these studies, it is reasonable to exclude nitrogen fixation as a relevant N source in the SIS and neglect it in the model. In the future, we will consider to include nitrogen fixation in the model if there are more observations available.

**Reference:**

Hashimoto, R., Watai, H., Miyahara, K., Sako, Y., and Yoshida, T.: Spatial and temporal variability of unicellular diazotrophic cyanobacteria in the eastern Seto Inland Sea, Fish Sci, 82, 459–471, https://doi.org/10.1007/s12562-016-0983-y, 2016.

Lee, Y. S., Seiki, T., Mukai, T., Takimoto, K., and Okada, M.: Limiting nutrients of phytoplankton community in Hiroshima Bay, Japan, Water Research, 30, 1490–1494, 1996.

Yamamoto, T., Hiraga, N., Takeshita, K., and Hashimoto, T.: An estimation of net ecosystem metabolism and net denitrification of the Seto Inland Sea, Japan, Ecological Modelling, 215, 55–68, https://doi.org/10.1016/j.ecolmodel.2008.02.034, 2008.

The sentences added to the revised manuscript are:
Added in Section 2.3:

"Furthermore, we also excluded nitrogen fixation as a nutrient source for SIS. Hashimoto et al. (2016) reported that nitrogen load from nitrogen fixation was much lower than nitrogen input from rivers in Osaka Bay and Lee et al. (1996) indicated that there was no nitrogen fixation in Hiroshima Bay. In a previous study estimating the nitrogen budget for the SIS, nitrogen fixation was assumed to be zero (Yamamoto et al., 2008)."

**RC7:** Section 4.2 is lacking information on how the figures presented in the article relate to previous estimates of the nitrogen budget of the SIS.

**AR7:** Thank you for your suggestions. We have collected some related information and will present it in this response note. We will also add this information in Section 4.2 of the revised manuscript.

Yanagi (1997) estimated the nitrogen budget in the SIS based on some assumptions. It reported that 392 mol N s$^{-1}$ of TN was transported from the land and 31 mol N s$^{-1}$ was deposited by rainwater. 358 mol N s$^{-1}$ of TN was transported to the open ocean at the open boundaries, and 64 mol N s$^{-1}$ was buried in the sediment. Our

study revealed that 64 mol N s$^{-1}$ of DIN was from rivers, among which 14 mol N s$^{-1}$ was transported at the open boundaries and 50 mol N s$^{-1}$ was buried in the sediment. Since we did not introduce the other types of nitrogen from rivers, our values are much lower than those reported by Yanagi (1997).

Fujiwara et al. (2006) also estimated the TN budget in the SIS and clarified the land origin and open ocean origin, showing that 330 mol N s$^{-1}$ of TN was supplied from the land to SIS, of which 297 mol N s$^{-1}$ was transported to the open ocean and 33 mol N s$^{-1}$ was buried to the sediment. Fujiwara et al. (2006) also reported that the net input of TN from the open ocean was 50 mol N s$^{-1}$, which was buried in the sediment. In this study, the TN originating from the open ocean has a net input of 62 mol N s$^{-1}$, all of which was buried in the sediment.

Compared our study with the above two studies, the main difference was the amount of TN from the land. Even though they made some adjustments for the TN obtained from the original unit method calculations, the estimates given based on experience have a high degree of uncertainty and were not linked to river discharges. Fujiwara et al. (2006) also stated that TN from the land they estimated had a great deal of uncertainty. According to our answer to RC2, we believe that it is more accurate to combine the river flow and the DIN load occurring on land to give the actual load flowing into SIS in the revised manuscript.

There are also studies to estimate the DIN transport at boundaries between the SIS and the open ocean. At the south of Bungo Channel (the west open boundary of our model), Morimoto et al. (2022) reported that a net of 385 mol N s$^{-1}$ oceanic DIN was transported from the open ocean to the SIS in July and August 2018 based on simulated water volume and DIN concentration derived from water temperature. It was 245 mol N s$^{-1}$ in our study. The reason that our estimate is less than theirs may be caused by the outward DIN transport. We used the DIN concentration calculated by the low-trophic ecosystem model, which was larger than their DIN concentration. Another possible reason is that the simulation of July and August 2018 by Morimoto et al. (2022) is for an extremely heavy rain case, which induced a stronger outflow in the surface layer and a stronger inflow in the bottom layer than those in our simulation. Fujiwara et al. (1997) reported about 168 mol N s$^{-1}$ of DIN through Kii Channel from the open ocean in August of 1985. We estimated about 139 mol N s$^{-1}$ of DIN was from the open ocean to the SIS through Kii Channel in August.

In revision, we added the above information to the manuscript.

**Reference:**

Fujiwara, T., Uno, N., Tada, M., Nakatsuji, K., Kasai, A., and Sakamoto, W.: Inflow of nitrogen and phosphorus from the ocean into the Seto Inland Sea, Proc. Coastal Engineering (JSCE), https://doi.org/10.2208/proce1989.44.1061, 1997.

Fujiwara, T., Kobayashi, S., Kunii, M., and Uno, N.: Nitrogen and phosphorus in Seto Inland Sea: Their origin, budget and variability, Bull Coast Oceanogr, 43, 129–136, 2006.

Morimoto, A., Dong, M., Kameda, M., Shibakawa, T., Hirai, M., Takejiri, K., Guo, X., and Takeoka, H.: Enhanced Cross-Shelf Exchange Between the Pacific Ocean and the Bungo Channel, Japan Related to a Heavy Rain Event, Front. Mar. Sci., 9, 869285, https://doi.org/10.3389/fmars.2022.869285, 2022.

Yanagi, T.: Budgets of fresh water, nitrogen and phosphorus in the Seto Inland Sea, Umi-no-Kenkyu, 6, 157–161, 1997.

The sentences added to the revised manuscript are:

Added in Section 3.5:

"At the south of Bungo Channel, Morimoto et al. (2022) reported that a net of 385 $mol\ s^{-1}$ oceanic DIN was transported from the open ocean to the SIS in July and August based on simulated water volume and DIN concentration derived from water temperature. It was 245 $mol\ s^{-1}$ in our study. The reason that our estimate is less than theirs may be caused by the outward DIN transport. We used the DIN concentration calculated by the hydrodynamic-biogeochemical model, which was larger than their DIN concentration. Another possible reason is that the simulation of July and August 2018 by Morimoto et al. (2022) is for an extremely heavy rain case, which induced a stronger outflow in the surface layer and a stronger inflow in the bottom layer than those in our simulation. Fujiwara et al. (1997) reported about 168 $mol\ s^{-1}$ of DIN through Kii Channel from the open ocean in August of 1985. Kasai et al. (2001) estimated the net DIN transport through Kii Channel into SIS in August of 1996 was 111 $mol\ s^{-1}$ which was similar to that of Fujiwara et al. (1997). We estimated about 150 $mol\ s^{-1}$ of DIN was from the open ocean to the SIS through Kii Channel in August."

Added in Section 4.2:

"There have been some studies that estimated the TN budget in the SIS based on certain assumptions. Yanagi (1997) reported that 392 $mol\ s^{-1}$ of TN was transported from the land, and 31 $mol\ s^{-1}$ from atmospheric deposition. Among these inputs, 358 $mol\ s^{-1}$ of TN was transported to the open ocean at the open boundaries, while 64 $mol\ s^{-1}$ was buried in the sediment. Our study revealed that 192 $mol\ s^{-1}$ of DIN was from rivers. After undergoing biogeochemical processes, 110 $mol\ s^{-1}$ was transported to the open ocean as sum of DIN, PHY, PON, and ZOO, while 82 $mol\ s^{-1}$ was buried in the sediment in the form of PON. It is important to consider a few factors when comparing our results to those in Yanagi (1997). First, we did not introduce the other types of nitrogen from the land. Additionally, the TN load used in Yanagi (1994) was obtained in 1982, a period characterized by severe terrestrial pollution. As a result, the nitrogen load from land in our study was lower compared to those reported by Yanagi (1997).

Fujiwara et al. (2006) also estimated the TN budget in the SIS and clarified the land origin and open ocean origin, showing that 330 $mol\ s^{-1}$ of TN was supplied from the land to SIS, of which 297 $mol\ s^{-1}$ was transported to the open ocean and 33 $mol\ s^{-1}$ was buried in the sediment. Fujiwara et al. (2006) also reported that the net input of TN from the open ocean was 50 $mol\ s^{-1}$, which was buried in the sediment. In our study, the TN originating from the open ocean has a net input of 61 $mol\ s^{-1}$, all of which were buried in the sediment."

**RC8:** Section 4.3 occurs very unexpectedly. If nutrient load reduction experiments are performed, this should be mentioned in the methods section and the results section and not appear for the first time in the discussion section. Anyway, the model with its assumed constant sedimentary N fluxes seems not appropriate for

nutrient load scenarios, since here the sediment feedback is essential. Your model implicitly assumes that as soon as some riverine N reaches the sediment in particulate form, its influence is gone. In reality, specifically in shallow near-coastal sediments, fresh organic matter that reaches the sediment can me remineralized quickly and (in case that this does not happen due to denitrification) become available for primary production again. So maybe leave just leave out this section (it adds a side-story to the main story line of the article) or move it to the online supplement?

**AR8:** Thank you for your suggestions. Yes, Section 4.3 is given a little unexpectedly. We will mention these sensitive experiments in the Methods section of the revised manuscript. The purpose of these sensitivity experiments is to examine the uncertainty of model results due to the change in the input flux of each source of nutrients. The processes you mentioned about the fresh matter that reaches the sediment and is remineralized quickly can be understood to be included in the nitrogen cycle within the water column in our model. This is because we introduce the resuspension processes in our model. If the bottom stress is over a critical value, the particles that reach the bottom will be returned to the water immediately. Then they will be remineralized quickly in the water column.

Again, this is also related to the definition of sediment source of nutrients. In our study, we do not want to treat such quickly remineralized nutrients as the sediment source. In our early calculation, we treated the quickly remineralized nutrients as the sediment source but found that the sediment source of nutrients became over 80% in most areas. Therefore, such bottom-touched particles were not allowed to be the sediment source of nutrients.

In the revision, we added some sentences to explain the above points.

The sentences added to the revised manuscript are:
Added in Section 2.5:

**2.5 Sensitivity experiments**

Understanding the response of SIS to the changes and uncertainty in nutrient inputs from different nutrient sources is crucial for effective nutrient management. To investigate this, we conducted sensitivity experiments by varying the input amount of each nutrient source individually. Nutrient inputs were altered by adding or subtracting the standard deviation of long-term variation of nutrient input based on the climatological input amount. This allowed us to simulate the responses of SIS to the larger and smaller inputs that may occur.

For the DIN load from rivers, we added twice the standard deviation to the climatological value for the upper limit but removed only one standard deviation from the climatological value for the lower limit to avoid negative values. The variation of DIN loads from rivers was specified in the model by changing the nutrient concentration in river water to avoid changes in the hydrodynamic fields due to the change in river discharge. For DIN load from the open ocean, the nutrient concentration at the open boundaries was added or minus one standard deviation to obtain the upper limit or lower limit, and the open boundary conditions for the hydrodynamic model were not changed. For DIN load from the sediment, TN concentration at the sediment surface in the 1980s was selected as the upper limit and TN concentration in the 2010s was selected as the

lower limit because the TN data in the 1980s, 1990s, 2000s, and 2010s show a reduction trend throughout these years. Table 1 summarized the input amount of DIN from the open ocean, rivers, and sediment in each sensitivity experiment.

**Table 1**. Annual mean input amount of DIN ($mol\ s^{-1}$) from the open ocean, rivers, and sediment in sensitivity experiments. "L" means the lower limit case; "U" means the upper limit case. The percentages mean the relative change from the value in the control case.

| Name of cases | From rivers | From open ocean | From sediment | Total input |
|---|---|---|---|---|
| Control | 192 | 799 | 86 | 1077 |
| L-open ocean | 192 | 527 (-34%) | 86 | 805 (-25%) |
| U-open ocean | 192 | 1073 (+34%) | 86 | 1351 (+25%) |
| L-rivers | 130 (-33%) | 799 | 86 | 1015 (-5.8%) |
| U-rivers | 314 (+64%) | 799 | 86 | 1199 (+11%) |
| L-sediment | 192 | 799 | 60 (+30%) | 1051 (-2.4%) |
| U-sediment | 192 | 799 | 136 (+58%) | 1127 (+4.6%) |

**Minor comments**

**RC9:** Line 30: "regulated" -> "influenced"? (Climate change has no "regulating" effect)

**AR9:** Agree. We corrected this in the revised manuscript.

**RC10:** Line 33:   "presenting a different seasonal variation" -> "so their import has a seasonality that is different."

**AR10:** Agree. We corrected this in the revised manuscript.

**RC11:** Line 56: Abbreviation "COD" is not defined.

**AR11:** We added its definition as "Chemical Oxygen Demand" in the revised manuscript.

**RC12:** Line 58: "concern about oligotrophication was raised for it" is unclear, please rephrase.

**AR12:** We rephrased this in the revised manuscript.

The revised expression is: "As a result of a long time of effort, the nutrient concentration in some areas of the SIS has largely decreased, which raised a social concern about the possibility of oligotrophication in the SIS (Yamamoto, 2003)."

**RC13:** Line 59: meaning of "As the first step" is unclear. Are you doing a multi-step approach, or do you indicate that you are the first who try to understand these changes?

**AR13:** We mean there are several steps to understanding the long-term change in the nutrient concentrations in the SIS. In this study, we conducted the climatological simulation to quantitatively evaluate the inventory of materials originating from the open ocean, river, and sediment. In the future study, we will conduct simulations for yearly and interannual variations to figure out the long-term variation of impacts of the open ocean, rivers, and sediment.

To avoid misunderstanding, we modified this sentence in the revised manuscript.

The revised expression is:

"To initiate our understanding of such long-term change in the nutrient concentrations in the SIS, …".

**RC14:** Line 91: "from a daily dataset" is too unspecific, please give a few more details.

**AR14:** We gave more details about the daily dataset in the revised manuscript.

The revised expression is:

"…from a daily dataset named Grid Point Value of Meso-Scale Model (GPV-MSM) provided by the Japan Meteorological Agency (http://www.jmbsc.or.jp/jp/online/file/f-online10200.html) (Zhu et al., 2019)."

**RC15:** Line 93: Please specify where your hydrodynamic boundary conditions come from.

**AR15:** We specified them in the revised manuscript.

The revised expression is:

"Four tidal constituents, $M_2$, $S_2$, $O_1$, and $K_1$, were specified at the open boundary of this model (Chang et al., 2009). In addition to the tidal components, the open boundary conditions also include the subtidal current velocity, temperature, and salinity from a diagnostic model (Guo et al., 2004).".

**RC16:** Line 112: "The spatial variation" -> "Spatial variation"

**AR16:** We corrected it in the revised manuscript.

**RC17:** Line 133: Wang 2002 actually only cites the method from Ariathurai and Krone (1976), please give the original reference.

**AR17:** We corrected it and add this literature in the Reference.

The revised expression is:

"…we followed the method proposed by Ariathurai and Krone (1976) and used Eq. (1):".

**RC18:** Line 172-177: Please state more clearly which fluxes you define at the boundaries. You state you define "zero concentration" but that is puzzling. At the land-sea and sediment-water boundaries you should have identical fluxes as for DIN for one of the tagged state variables and zero flux for the others. For the open boundary condition, this should be the same during times of inflow, but during times of outflow (in the upwind scheme) the DIN_??? should be exported according to the ratio DIN_???/DIN. Please clarify.

**AR18:** We stated more clearly the fluxes at the boundaries in the revised manuscript. Because we solve the DIN of each source, we have their value at the grid next to the open boundary. Therefore, we do not need to use the ratio of DIN_???/DIN to determine the flux for outflow.

The revised expression is:

"For the open boundary conditions, during the time of inflow, $DIN_{ocean}$ flux had the same values as those used at the open boundaries of the hydrodynamic-biogeochemical model; during the time of outflow, $DIN_{ocean}$ flux was given as the product of $DIN_{ocean}$ and the outflow velocity. $DIN_{ocean}$ flux was specified to zero at the land-sea interface and the water-sediment interface. $DIN_{river}$ at the land-sea interface was identical to those used in the hydrodynamic-biogeochemical model, but it was set to zero at the water-sediment interface. At the open boundaries, during the time of inflow, $DIN_{river}$ flux was set to zero and during the time of outflow, $DIN_{river}$ flux was given as the product of $DIN_{river}$ and the outflow velocity. $DIN_{sediment}$ was set to have the same flux at the sediment-water interface as that in the hydrodynamic-biogeochemical model, but it was set to have zero flux at the land-sea interface. At the open boundaries, during the time of inflow, $DIN_{sediment}$ flux was set to zero and during the time of outflow, $DIN_{sediment}$ flux was given as the product of $DIN_{sediment}$ and the outflow velocity."

**RC19:** Line 186: Why do you use observations in 50 m depth as "bottom value" for areas deeper than 50 m? Please clarify.

**AR19:** According to the report released by the Ministry of the Environment, Japan (https://www.env.go.jp/content/900530598.pdf), it explained that the original data for those stations deeper than 50 m were sampled at 50 m in the Broad Comprehensive Water Quality Survey.

In order to avoid misunderstanding the definition of the bottom layer, we modified the related sentences in the revised manuscript.

The revised expression is:

"At each station, the data were sampled from two layers, the upper layer, located 1 m below the sea surface, and the lower layer, positioned 1 m above the sea floor for stations shallower than 50 m. For stations deeper than 50 m, the data for the lower layer were obtained at a fixed depth of 50 m (https://www.env.go.jp/content/900530598.pdf)."

**RC20:** Section 3.1: While Fig. 2 and Fig. 3 are good for showing how well the model captures the spatial signal, it is really hard to see by eye whether it also resolves the seasonal patterns. I suggest adding a few

climatologies from the model compared to observations, for a few stations representative for different subareas of the model domain. This is probably sufficient in the supplement.

**AR20:** Thank you for your suggestions. The observation data from the Ministry of the Environment, Japan covers the whole SIS, and the sampling date in January, May, July, and October, representing winter, spring, summer, and autumn. We calculated the simulated DIN and PHY concentration of monthly mean in each sub-region of SIS and the comparison with the observations was provided in the Supplement Materials of the revised manuscript.

Added in Supplement Materials:

[Figure]

**Figure S6.** Monthly mean observed and simulated DIN ($mmol\ m^{-3}$) for the upper and lower years of each subregion of SIS. The observation stations in each subregion are presented in Fig. 1(c).

[Figure]

**Figure S7.** Monthly mean observed and simulated PHY concentration ($mg\ Chla\ m^{-3}$) for the upper and lower years of each subregion of SIS. The observation stations in each subregion are presented in Fig. 1(c).

**RC21:** Line 234-238: "have already occupied most areas of the SIS": it would be better to calculate the ratio (DIN_ocean+DIN_river+DIN_sediment)/DIN. If that is close to one everywhere in the model domain, you can estimate that your spin-up period for the tagging is completed.

**AR21:** We added one figure to describe the ratio of (DIN_ocean+DIN_river+DIN_sediment)/DIN in the SIS from the first year to the third year of the tracking simulation in the Supplement Materials of the revised manuscript.

Added in Supplement Materials:

[Figure]

**Figure S8. (a-r)** The monthly mean ratio of water column averaged $(DIN_{ocean} + DIN_{river} + DIN_{sediment})/DIN$ over the first three years. **(s)** The daily mean ratio of volume averaged $(DIN_{ocean} + DIN_{river} + DIN_{sediment})/DIN$ over the first three years.

**RC22:** Line 373: "whose ratio is 1.4:1": The ratio between what? Subsequently more occurrences.

**AR22:** In the original manuscript, this ratio is between the horizontal export flux of biological particles (PHY+ZOO+PON) to the open ocean (187 mol N s$^{-1}$) and the vertical export flux of biological particles to the sediment (136 mol N s$^{-1}$).

187 mol N s$^{-1}$/ 136 mol N s$^{-1}$≈1.4

We gave more explanations in the revised manuscript.

The revised expression is:

"In the SIS, the horizontal export flux of biological particles (PHY+ZOO+PON) to the open ocean is 229 $mol\ s^{-1}$ (Fig. 10a) and the vertical export flux of biological particles to the sediment is 190 $mol\ s^{-1}$, whose ratio is 229 $mol\ s^{-1}$/190 $mol\ s^{-1}$ ≈1.2:1. If we examine them for the different origins of nutrients, this ratio changes. For oceanic nutrients (Fig. 10b), the horizontal export of biological particles has a flux of 140 $mol\ s^{-1}$ while the vertical export has a value of 61 $mol\ s^{-1}$, whose ratio is 140 $mol\ s^{-1}$/61 $mol\ s^{-1}$≈2.3:1; for the riverine nutrients (Fig. 10c), the horizontal export has a flux of 68 $mol\ s^{-1}$ while the vertical export has a value of 82 $mol\ s^{-1}$, whose ratio is 68 $mol\ s^{-1}$/82 $mol\ s^{-1}$≈0.83:1; for the benthic nutrients (Fig. 10d), the horizontal export has a flux of 21 $mol\ s^{-1}$ while the vertical export has a value of 47 mol s$^{-1}$, whose ratio is 21 $mol\ s^{-1}$/47 $mol\ s^{-1}$≈0.45:1."

**RC23:** Line 454: "the management can also be applied to the sediments": It is very unclear how you would "manage" sedimentary nutrient release, you cannot easily modify it. If this is a serious option please give more details, e.g. will you add substances to capture some of the escaping nutrients?

**AR23:** Thank you for your comment. We intend to raise awareness about the sediment release in the SIS. As you stated, the expression "the management can also be applied to the sediments" is not appropriate. We modify this sentence in the revised manuscript.

The revised expression is:

"…it needed to pay more attention on the sediments".